# Modeling the Coupled and Decoupled states of Polar Boundary-Layer Mixed-Phase Clouds

Étienne Vignon[*,1,3], Lea Raillard[*,1], Audran Borella[2], Gwendal Rivière[1], and Jean-Baptiste Madeleine[1]

[*]These authors contributed equally to this work.
[1]Laboratoire de Météorologie Dynamique-IPSL, Sorbonne Université/CNRS/ Ecole Normale Supérieure-PSL Université/ Ecole Polytechnique-Institut Polytechnique de Paris, Paris, France
[2]Institut Pierre-Simon Laplace, Sorbonne Université/CNRS, Paris, France
[3]Laboratoire de Physique et Chimie de l'Environnement et de l'Espace (LPC2E), Université d'Orléans, CNRS UMR7328, CNES, Orléans, France

**Correspondence:** Étienne Vignon (etienne.vignon@lmd.ipsl.fr)

**Abstract.** Representing mixed-phase clouds (MPCs) is a long-standing challenge for climate models, with major consequences regarding the simulation of radiative fluxes at high-latitudes and uncertainties in future cryosphere melting estimates. Low-level boundary-layer MPCs that prevail at high-latitudes can be either coupled or decoupled to the surface, which modulates their dynamical and microphysical properties. This study leverages a recent physically-based parameterization of phase partitioning considering an explicit coupling between microphysics and subgrid-scale dynamics and involving direct interactions between the cloud and turbulent diffusion schemes. This parameterization makes it possible to capture the structure of the decoupled state of polar boundary-layer MPCs – with a supercooled liquid dominated cloud-top sitting on top of precipitating ice crystals – in single column simulations with the LMDZ Atmospheric General Circulation Model. The positive feedback loop involving cloud-top radiative cooling induced by supercooled liquid droplets, subsequent buoyancy production of turbulence as well as the supercooled liquid water production associated with turbulence, is captured by the model. However, the liquid and cloud ice water path remain underestimated and most of the turbulence is confined near cloud top which is probably due to a missing parameterization for convective downdrafts in the model. The study further shows that accounting for the detrainment of shallow convective plume's air when diagnosing the in-cloud supersaturation makes it possible to capture the overall vertical structure of surface-coupled clouds, with realistic liquid and ice water contents. Nonetheless, a parameteric sensitivity analysis emphasizes the importance of properly calibrating the parameter controling the supercooled liquid water production term by subgrid turbulence.

## 1   Introduction

The representation of mixed-phase clouds (MPCs) is a long-standing challenge for numerical weather prediction and climate models (Korolev et al., 2017; Furtado, 2018). Low-level MPCs have been shown to be the source of strong biases in radiative fluxes and surface temperature over the Southern Ocean in climate models involved in the Coupled Model Intercomparison Projects (CMIP) (Bodas-Salcedo et al., 2014, 2016; Hyder et al., 2018; Cesana et al., 2022). Those biases can be partly

corrected with a targetted tuning (e.g, Hourdin et al., 2020) but this often hides error compensations since current subgrid cloud parameterizations fail to properly represent the physics of MPCs (e.g, Forbes and Ahlgrimm, 2014; Lenaerts et al., 2017; Furtado, 2018; Vignon et al., 2021). Importantly, the equilibrium climate sensitivity of CMIP models is very sensitive
to the liquid water content simulated in austral MPCs (Lohmann and Neubauer, 2018; Gettelman et al., 2019; Zelinka et al., 2020; Cesana et al., 2024) as supercooled liquid water (SLW) droplets substantially enhance the cloud albedo and therefore the amount of shortwave radiation reflected back to space and at the same time, they significantly increase the cloud optical depth and the cloud radiative forcing in the infrared spectrum. Low-level MPCs also play an important role in the Arctic climate system through interactions with sea ice (Kay et al., 2016a), the atmospheric boundary layer (Pithan et al., 2016)
and boreal continental surfaces (Cronin and Tziperman, 2015). An accurate representation of MPCs in atmospheric models is thus paramount to understand and accurately simulate polar climate (Engström et al., 2014; Lawson and Gettelman, 2014) as well as understand and anticipate the contribution of cloud feedbacks to polar amplification (Tan and Storelvmo, 2019). Moreover, the variability in model projections of snow and ice melting over the Greenland ice-sheet and Antarctic ice shelves primarily depends upon the differences in MPCs' SLW content simulated by the models. These differences strongly influence
the longwave cloud radiative effect and, in turn, the surface radiative budget (Hofer et al., 2019; Kittel et al., 2022). An accurate representation of MPCs in models is therefore critical to simulate the high-latitude and global climate systems and to reduce uncertainty in climate projections.

Observational campaigns at the poles have revealed the resilience of boundary-layer MPCs which can persist for several days. This resilience could *a priori* be surprising given the thermodynamical instability of SLW droplets at $T < 0°C$ and their
depletion through vapor deposition towards ice crystals as the relative humidity with respect to ice exceeds 100 % (Shupe et al., 2006; Morrison et al., 2012). The formation of SLW in polar boundary-layer clouds results from interactions between turbulence, microphysics and radiation (Korolev et al., 2017). In turbulent updrafts, generated either by convective instability at the surface (Shupe et al., 2008) or by cloud-top eddies induced by radiative cooling (Simpfendoerfer et al., 2019; Barrett et al., 2020), the relative humidity can reach saturation with respect to liquid through air adiabatic cooling during ascent (Korolev
and Mazin, 2003). Cloud droplets can thus form almost adiabatically and are advected upward, thereby forming a thin – a few hundred meters deep – liquid layer. Most of the time, the scarcity of ice nucleating particles (INPs) (Eirund et al., 2019; Creamean et al., 2022; Wex et al., 2025) in polar regions makes heterogeneous freezing processes weakly active. Subsequently, the vapor deposition overall serves as a very weak sink of moisture (Silber et al., 2021) which explains the commonality and resilience of liquid-bearing clouds (Silber et al., 2020) at the poles. The growth of ice crystals through vapour deposition and
riming (Maherndl et al., 2024; Chellini and Kneifel, 2024) make them sediment below – and separate from – the liquid layer.

Polar boundary-layer MPCs, in particular stratocumulus clouds that prevail at those latitudes, can be either coupled or decoupled to the sea, land or ice surface (Shupe et al., 2013). The coupling occurs when the turbulent cloud associated-mixed layer extends down to the surface-based boundary layer. The coupling state can affect the microphysical properties of the clouds (Gierens et al., 2020). For example, the transport of marine INPs from the surface up to the cloud is favored in the coupled
state (Griesche et al., 2021). While coupled MPCs most often form above open ocean at the top of convective boundary layers, decoupling can occur when these clouds are advected over stably stratified near-surface air above sea ice (Pithan et al., 2018)

or ice shelves. Decoupling can be further enhanced by the sublimation of sub-cloud ice precipitation leading to local cooling and creation of sub-cloud stratification (Sotiropoulou et al., 2014). Surface-decoupled convective MPCs can also form through radiative cooling at the top of stratus or fog clouds in a stratified environnement (Simpfendoerfer et al., 2019). When the cooling is sufficiently intense, a mixed-layer develops within and below the cloud through cloud-top convective instability.

Some large eddy simulation (LES) models, cloud-resolving models and mesoscale models can reasonably capture the structure of both surface coupled and surface-decoupled boundary-layer MPCs as long as ice formation rate and ice properties are realistic (e.g., Klein et al., 2009; Ovchinnikov et al., 2014; Arteaga et al., 2024; Silber et al., 2019; Tornow et al., 2025). However, General Circulation Models (GCMs) still struggle to simulate the vertical structure and microphysical properties of surface-coupled clouds (e.g., Liu et al., 2011; Gettelman and Morrison, 2015; Zhang et al., 2020). These shortcomings in GCMs are even more pronounced for surface-decoupled MPCs, even though recent single-column simulations with the NASA ModelE3 model show promising results, including onset of turbulence from a purely liquid stratiform cloud and subsequent triggering of ice precipitation (Silber et al., 2022). Overall, shortcomings in the representation of polar boundary-layer MPCs in GCMs lead to substantial biases in the representation of the surface-based temperature inversion over the wintertime Arctic sea ice in cloudy conditions (Pithan et al., 2014).

Overall, the parameterization of MPCs in GCMs remains extremely challenging and the difficulty mostly lies in: (i) the parameterization of ice microphysical processes (Forbes and Ahlgrimm, 2014; Barrett et al., 2017a; Vignon et al., 2021), in particular the parameterizations of ice formation rate in the immersion mode (e.g., Knopf et al., 2023) and of secondary ice production processes (e.g., Pasquier et al., 2022; Sotiropoulou et al., 2020; Possner et al., 2024); (ii) a missing or insufficient coupling between the cloud condensation scheme (or microphysics scheme) and the turbulent mixing and shallow convection parameterizations which precludes a direct dependence of the cloud liquid water content to subgrid vertical motions and turbulence (Storelvmo et al., 2008; Field et al., 2014; Furtado et al., 2016); (iii) a poor vertical resolution which prevents from representing the vertical profiles of temperature, moisture and cloud water content near cloud top (Barrett et al., 2017b) and ; (iv) the assumptions made regarding the subgrid spatial mixing between ice crystals and supercooled droplets (Rotstayn et al., 2000; Korolev and Milbrandt, 2022) which impacts on the treatment of microphysical processes.

In this paper, we show that a physically-based parameterization of MPCs considering an explicit coupling between microphysics and subgrid-scale dynamics and involving direct interactions between the cloud scheme, the turbulent diffusion scheme, and the shallow-convection scheme, makes it possible to capture the general structure of both coupled and decoupled states of polar boundary-layer MPCs in single column simulations with the LMDZ GCM.

The article is structured as follows. Section 2 describes the LMDZ GCM, its cloud parameterization as well as the two single column simulation setups. Section 3 presents the results of the analysis of single column simulations of both coupled and decoupled MPCs. Section 4 closes the paper with a conclusion.

## 2 Model and simulation set-up

### 2.1 The LMDZ GCM and the different parameterizations of MPCs

#### 2.1.1 LMDZ GCM and default parameterization of MPCs

LMDZ is the global atmospheric component of the IPSL-CM Earth System Model (Boucher et al., 2020), historically and still actively involved in the CMIP exercises and named after the French climate institute where it is developed: the Institut Pierre-Simon Laplace (IPSL). A recent set of papers (Hourdin et al., 2020; Cheruy et al., 2020; Madeleine et al., 2020) describes the recent developments and the performances of the 6th version of LMDZ used for CMIP6 that we consider in this study and briefly describe hereafter.

The turbulent vertical flux $\overline{\rho w' \psi'}$ of a variable $\psi$ in the boundary-layer is parameterized with an Eddy Diffusivity - Mass Flux approach and reads:

$$\overline{\rho w' \psi'} = -K_\psi \frac{\partial \psi}{\partial z} + f_{th}(\psi_{th} - \psi) \tag{1}$$

where $w'$ and $\psi'$ are the turbulent fluctuations of the vertical velocity and of $\psi$ respectively, the overline is the average operator in the Reynolds' equations system framework. The first term on the right hand side is the counter-gradient diffusion with $K_\psi$ the eddy diffusivity calculated with a TKE-l turbulent scheme. It is worth noting that the TKE-l scheme has recently been updated and we use here the new scheme from Vignon et al. (2024) that exhibits better numerical properties as well as more robust and more easily tunable formulations of the different terms of the eddy diffusivity coefficients compared to the previous TKE-l scheme of the model (Vignon et al., 2017). The second term on the right hand side of Eq. 1 is the transport of $\psi$ by the so-called 'Thermal Plume Model' (Rio and Hourdin, 2008; Hourdin et al., 2019), a mass-flux scheme that summarizes the mean behaviour of a population of shallow-convective plumes and rolls. Each atmospheric column is divided into a mean ascending updraft of mass flux $f_{th}$ and the 'environment' i.e. the region with a compensating downdraft. $\psi_{th}$ is the value of $\psi$ within thermals. $f_{th}$ can be estimated from the continuity equation:

$$\frac{\partial f_{th}}{\partial z} = e_{th} - d_{th} \tag{2}$$

where $e_{th}$ and $d_{th}$ are the lateral entrainement (resp. detrainment) rate of air towards (resp. away from) the thermals. Further elements on $e_{th}$ and $d_{th}$ calculation are given in Rio et al. (2010). Importantly, this mass-flux scheme only activates from the ground surface, when surface convective instability occurs. The combination of the LMDZ eddy-diffusivity and mass-flux schemes has proven successful in representing the structure of stratocumulus clouds, and in particular, the cloud-top dynamics (Hourdin et al., 2019).

The parameterization of clouds in LMDZ follows a purely macrophysical approach extensively described in Madeleine et al. (2020). It is based on a statistical scheme that assumes a subgrid distribution of the total water vapour from which the cloud cover and the cloud total water specific content are diagnosed. In absence of deep or shallow convection from the surface –

conditions under which surface-decoupled MPCs are found – the LMDZ cloud parameterization considers a subgrid Probability Density Function (PDF) $F(q)$ of the total water content $q$. This PDF follows a log-normal distribution bounded by 0 (Bony and Emanuel, 2001, right column in Figure 1). Assuming instantaneous adjustment to saturation, the cloud fraction $\alpha_c$ and the cloud total water – sum of vapor and condensates – specific content $q^{cld}$ (kg kg$^{-1}$) read:

$$\alpha_c = \int_{q_{lim}}^{\infty} F(q)\, dq \tag{3}$$

$$q^{cld} = \int_{q_{lim}}^{\infty} q\, F(q)\, dq. \tag{4}$$

The condensation threshold $q_{lim}$ is taken equal to the saturation specific humidity with respect to liquid $q_{sl}$ (resp. with respect to ice $q_{si}$) when $T \geq 0°C$ (resp. $T < 0°C$).

In presence of surface-based shallow convection, i.e. where the Thermal Plume Model is active, cloud fraction and specific cloud water content are calculated from a bi-gaussian PDF (left column in Figure 1) of the saturation deficit $s$. It is defined as $s = a_l(q - q_{lim}(T_l))$ where $T_l$ is the liquid temperature and $a_l$ a thermodynamic function that depends upon temperature (Jam et al., 2013). Again, $q_{lim}$ is equal to the saturation specific humidity with respect to liquid $q_{sl}$ (resp. with respect to ice $q_{si}$) when $T \geq 0°C$ (resp. $T < 0°C$). One gaussian distribution is associated with the air properties within thermals, the other one with the properties of the air within the environment. Each of them is characterized by the mean saturation deficit ($s_{th}$ or $s_{env}$) provided by the Thermal Plume Model and a standard deviation ($\sigma_{th}$ or $\sigma_{env}$). $\sigma_{th}$ and $\sigma_{env}$ are determined by considering that the width of the distributions of $s$ in thermals and the environment are mostly driven by the exchange of air between the plume and the environment. The dispersion of $s$ in each of the two regions therefore increases when $s_{th} - s_{env}$ increases (see Jam et al., 2013 and Hourdin et al., 2019 for details). Cloud fraction and cloud total water content are also computed assuming instantaneous adjustment to saturation in each of the two regions, the condensation threshold being $s = 0$ which corresponds to saturation in each region.

Assuming that the specific content of vapour $q_v = q_{lim}$ in all types of clouds, the mass content of cloud condensates is then partitioned between the liquid and ice phases by computing the fraction of cloud condensed water in the liquid-phase $x_{liq}$ using a continuous function of temperature $T$ (Madeleine et al., 2020) :

$$x_{liq}(T) = \frac{q_l}{q_l + q_i} = \left( \frac{T - T_{min}}{T_{max} - T_{min}} \right)^{0.5}, \tag{5}$$

where $q_l$ and $q_i$ are the specific contents of liquid droplets and ice crystals respectively, $T_{min} = -30°C$ and $T_{max} = 0°C$.

The treatment of cold precipitation consists in diagnosing a vertical profile of precipitation flux assuming stationarity and equilibrium of the precipitation with the cloud field. Autoconversion of ice crystals is based on a sedimentation law and supercooled rain immediately freezes (see Section 2.7.3 of Madeleine et al. (2020)). Importantly, the specific contents of snow and rain are not variables of the scheme as the latter computes fluxes. Following Raillard et al. (2024), the snow specific content can be estimated offline from the snow flux, assuming a fixed fall velocity $v_i$ for ice crystals.

### 2.1.2 Recent advances in the parameterization of phase partitioning through the coupling with turbulent diffusion

In agreement with previous studies (Pithan et al., 2016; Forbes and Ahlgrimm, 2014), Raillard et al. (2024) showed that the
temperature-based cloud phase partitioning is inappropriate to simulate the structure of polar MPCs, particularly because SLW
droplets are often located near cloud top i.e. in the coldest part of the cloud. Subsequently, Raillard et al. (2025) developed a
phase partitioning parameterization based on the works of Field et al. (2014) and Furtado et al. (2016) that we briefly summarize
here. The general rationale of the parameterization stems from the conceptual physical model of Korolev and Mazin (2003)
that predicts the SLW production in vertically moving air parcels containing – or not – pre-existing ice crystals. It starts from a
slightly adapted version of the so-called linearized Squires' equation that predicts the evolution of supersaturation (here taken
with respect to the ice phase $S_i$) and reads :

$$\frac{dS_i}{dt} = -\underbrace{\frac{S_i}{\tau_p}}_{1} + \underbrace{a_i(\overline{w} + w')}_{2} \tag{6}$$

Term 1 is the relaxation of supersaturation due to vapour deposition upon pre-existing ice crystals, $\tau_p$ being the associated
timescale which is a function of temperature and of the first moment of the ice crystals' size distribution. Term 2 is the
source/sink term of supersaturation associated with vertical velocity that we arbitrarily decompose here into a large scale and
resolved component $\overline{w}$ and a turbulent subgrid one $w'$. Following Field et al. (2014), the $w'$ turbulent fluctuations are treated as
a white noise process and Eq. 6 becomes a stochastic differential equation which admits a steady-state solution with the form
of a Gaussian distribution $f(S_i)$ of variance $\sigma_s^2$ and mean $\langle S_i \rangle$.

$$f(S_i) = \frac{1}{\sqrt{2\pi\sigma_s^2}} \exp\left(-\frac{(S_i - \langle S_i \rangle)^2}{2\sigma_s^2}\right) \tag{7}$$

$$\sigma_s^2 = \frac{1}{2} a_i^2 \sigma_{w'}^2 \tau_d \tau_p \tag{8}$$

$$\langle S_i \rangle = a_i \overline{w} \tau_p. \tag{9}$$

The timescale $\tau_p$ is a function of the first moment of the ice crystal size distribution $\mathcal{M}_1$. The latter is calculated assuming an
exponential size distribution for ice crystals and using the pre-existing ice crystal mass concentration (at the beginning of the
model physics time step). We further assume that the ice crystal number concentration follows the number of ice nucleating
particles (INP) predicted by the DeMott et al. (2010) parameterization. The latter depends upon temperature and the number
concentration of aerosol particles with diameters larger than $0.5\ \mu m$ ($N_{aero5}$) that we fix in the simulations (see Section 2.2.2
of Raillard et al. (2025)).

The time-scale $\tau_d$ involved in Eq. 8 is the Lagrangian turbulent decorrelation time-scale (Rodean, 1996) which is the
characteristic timescale over which a turbulent flow loses memory of its initial state. Assuming near-isotropic turbulence,
the distribution of the turbulent vertical velocity of variance $\sigma_{w'}^2$ is related to the turbulent kinetic energy $e$ predicted by the
TKE-l turbulent diffusion scheme:

$$e = \frac{3}{2}\sigma_{w'}^2 \tag{10}$$

$\tau_d$ then reads:

$$\tau_d = \frac{4}{3}\frac{e}{\varepsilon C_0}\gamma_{\tau_d} \tag{11}$$

where $\varepsilon$ is the eddy dissipation rate, $C_0$ is the Lagrangian structure-function experimental constant and $\gamma_{\tau_d}$ a tuning coefficient.

Assuming a liquid saturation adjustment process, the liquid cloud fraction $\alpha_l$ and the specific amount of liquid $q_l$ are estimated by integrating the distribution from $S_{iw}$ the value of the supersaturation at the liquid water saturation point:

$$\alpha_l = \int_{S_{iw}}^{\infty} f(S_i)\,dS_i \tag{12}$$

$$q_l = \int_{S_{iw}}^{\infty} q_{si}(S_i - S_{iw})f(S_i)\,dS_i. \tag{13}$$

The liquid cloud fraction $\alpha_l$ is bounded such that $q_l$ does not exceed the total cloud condensed water content $q^{cld}$ - $q_{lim}$. $q_i$ is then estimated as the difference between the total cloud condensed water content and the estimated $q_l$.

In the LMDZ cloud scheme, the condensation procedure is iterative and the condensation threshold $q_{lim}$ is adjusted at each iteration. In the mixed-phase temperature regime, following Dietlicher et al. (2018), Raillard et al. (2025) made $q_{lim}$ depend on $x_{liq}$ at each iteration such that:

$$q_{lim} = x_{liq}\,q_{sl} + (1 - x_{liq})\,q_{si} \tag{14}$$

This enables more consistency between the liquid and ice water contents estimated from the cloud-phase partitioning scheme, and the cloud fraction $\alpha_c$ and total cloud water content $q^{cld}$ estimated from the condensation scheme. The prediction of the cloud liquid ratio $x_{liq}$ is thus no longer a function of temperature but depends on subgrid turbulent activity and the properties of ice crystals. Raillard et al. (2025) developed this parameterization for mid-level stratiform clouds and the authors kept the default temperature-dependent phase partitioning for convective boundary layer clouds namely for clouds formed through condensation using the bi-gaussian distribution of the saturation deficit. The rationale and hypotheses of Raillard et al. (2025)'s parameterization – and in particular the use of steady-state solution which is – make it *a priori* designed to capture the phase of stratiform and surface-decoupled boundary-layer MPCs. However, some adaptations are required to make it suitable for surface-coupled MPCs forming at the top of convective boundary layers.

### 2.1.3 Adaptations to capture SLW production in convective boundary layer clouds through the coupling with shallow convection

In convective boundary layer MPCs, in-cloud supersaturation and SLW production are strongly modulated by entrainment and detrainment to and from thermal plumes (Kay et al., 2016b). Entrainment is the mixing of drier environmental air into ascending thermal plumes, leading to a dilution of moisture and a subsequent reduction in supersaturation within the plume. Importantly, detrainment transfers moister air from the plumes into the surrounding environment, increasing local moisture content and

potentially generating supersaturation. This interaction between coherent turbulent updrafts and clouds in the environment should be distinguished from the local SLW generation associated with local, homogeneous, and nearly isotropic small-scale turbulent eddies considered in the previous parameterization. To account for this effect on clouds located in the environment surrounding convective thermal plumes, the Raillard et al. (2025)'s parameterization has been modified as follows. In cases for which the 'Thermal Plume Model' mass-flux scheme is active and for which clouds form using the bi-gaussian distribution of subgrid saturation deficit, we consider an adaptation of Eq. 6 with a 3rd 'homogeneisation' term:

$$\frac{dS_i}{dt} = - \underbrace{\frac{S_i}{\tau_p}}_{1} + \underbrace{a_i(\overline{w} + w')}_{2} - \underbrace{\frac{S_i - S_E}{\tau_E}}_{3}, \tag{15}$$

Such a term was originally introduced in Furtado et al. (2016) to account for the exchange of vertically moving air parcels with their surrounding whose supersaturation is $S_E$, with a typical timescale $\tau_E$. It was not taken into account in Raillard et al. (2025) in a first instance (see their motivations in their Sect. 2.2.1) but re-introduced here to account for air parcels mixing between clouds in the environment and the air in the thermals. $S_E$ is taken equal to the mean supersaturation within thermals (evaluated at the underlying layer such as it is not affected by the mixing with the environment at the same layer yet) and $\tau_E$ is taken inversely proportional to the magnitude of the net detrainment rate:

$$\tau_E^{-1} = \gamma_E \frac{\max(d_{th} - e_{th}, 0.)}{\rho} \tag{16}$$

where $\rho$ is the air density and $\gamma_E$ a tuning coefficient.

Eqs. 8 and 9 are then adapted to take into account the new term:

$$\sigma_s^2 = \frac{1}{2} \frac{a_i^2 \sigma_{w'}^2 \tau_d}{\frac{1}{\tau_p} + \frac{1}{\tau_E}} \tag{17}$$

$$\langle S_i \rangle = \frac{(a_i \overline{w} + S_E/\tau_E)}{\frac{1}{\tau_p} + \frac{1}{\tau_E}} \tag{18}$$

Figure 1 summarizes with schematics how the adapted LMDZ cloud scheme works when the new parameterization for phase partitioning is activated, considering separately surface-coupled – that is, when shallow convection is active – and surface-decoupled clouds.

## 2.2 LMDZ SCM simulations on the ISDAC and M-PACE cases

The performance of LMDZ to simulate surface-coupled and decoupled boundary-layer MPCs is assessed using Single Column Model (SCM) simulations of two contrasted case studies spanning different ranges of liquid and ice water path, namely the M-PACE and ISDAC cases (Fridlind and Ackerman, 2018). Those reference case studies have been extensively used to assess the performance of atmospheric models, from LES models to SCM configuration of GCMs, to simulate boundary-layer MPCs.

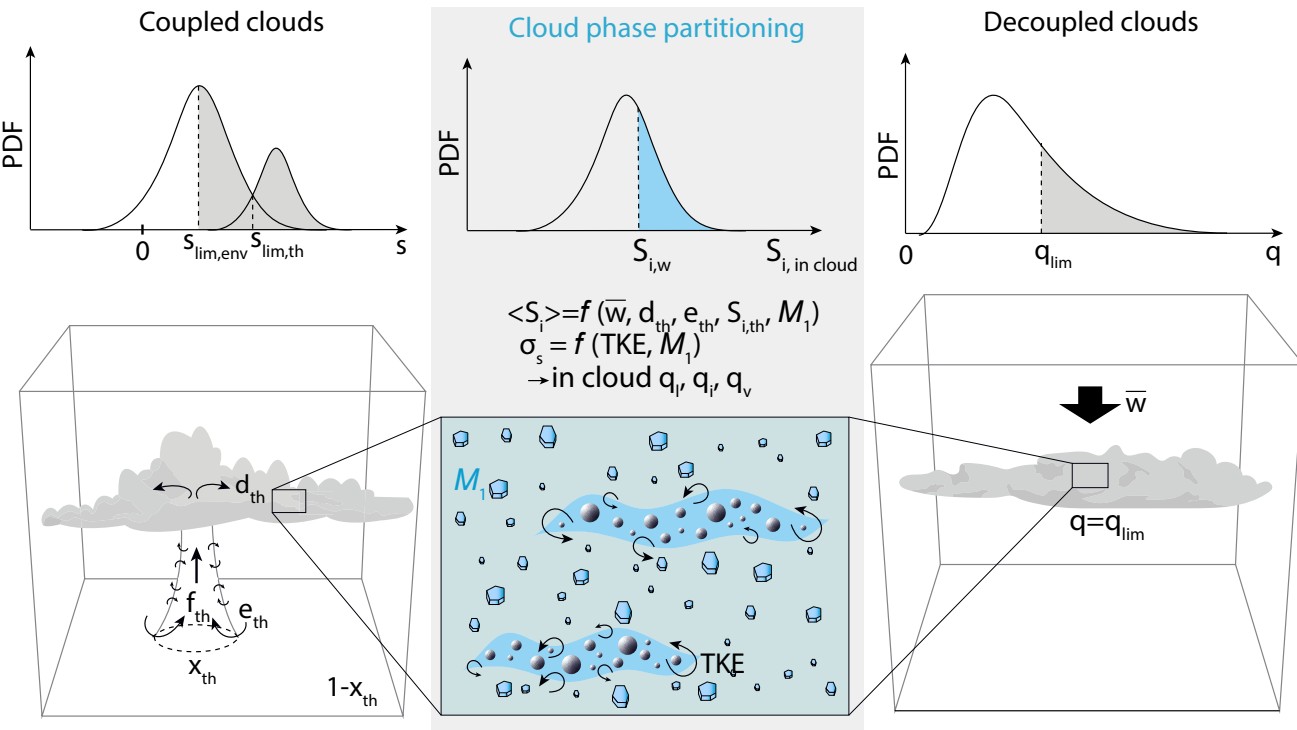

**Figure 1.** Schematics of LMDZ cloud parameterization and its coupling with the turbulent diffusion and shallow-convection scheme to simulate the phase of polar boundary-layer MPCs. See main text for notations.

### 2.2.1 The M-PACE case

The first case study focuses on a widespread single-layer cloud deck sampled during the Atmospheric Radiation Measurement Program's (ARM) Mixed-Phase Arctic Cloud Experiment (M-PACE, Verlinde et al., 2007) between 9 and 10 October 2004

along the North Slope of Alaska. Cloud formation was governed by strong sensible and latent surface heat fluxes during a cold-air outbreak event, where air of the $\approx$ 1500 m deep boundary layer was advected within a north-easterly flow from the coast of snow-covered Alaska to the open ocean surface under a persistent high pressure system. The case was built for a SCM and LES intercomparison exercise in Klein et al. (2009). The simulation is 12 h long and the initial state provided by radiosoundings analysis at Utqiagvik (formerly known as Barrow), Alaska, corresponds to a convective boundary layer with

purely liquid clouds topped by a 2 K temperature inversion. Surface temperature, turbulent sensible and latent heat fluxes are kept constant throughout the simulation. The advective forcings of the SCM are prescribed and consist in vertical profiles of temperature and humidity horizontal advection terms as well as that of vertical velocity that are constant in time. During the case study period, the cloud deck was sampled by 2 spiral flights of the North Dakota Citation aircraft. Here we focus on the observed liquid and ice water contents from the 1009 and 1010 flights and details on in-situ cloud probes's data process are

provided in McFarquhar et al. (2007). From airborne and ground-based remote sensing data during the case study, Klein et al. (2009) estimate the liquid water path (LWP) to range between 110 and 210 g m$^{-2}$ and the ice water path (IWP) between 8 and 30 g m$^{-2}$. The observed ice crystals number concentration throughout the cloud layer ranges between 0.1 and 10 L$^{-1}$ with most values of around 1 L$^{-1}$ (McFarquhar et al., 2007). The INP parameterization used in our cloud scheme with a default $N_{aero5}$ value of 1 scm$^{-3}$ provides an ice number concentration roughly between 0.1 and 0.6 L$^{-1}$ for the temperature range

within the M-PACE cloud layer, namely between $\approx$ -15 and -10°C. Those values are thus realistic but in the lower part of the measured range. Simulations with higher ice crystals number concentration will be investigated by varying N$_{aero5}$. M-PACE simulations are run with the standard 10-min time step of the LMDZ SCM, but additional simulations using a shorter 1-min time step reveal an overall weak sensitivity of the results (not shown here). Given the relatively high SLW content in clouds, precipitating ice crystals during M-PACE mostly correspond to rimed particles, mostly with irregular, rosette and columnar

shapes in the below-cloud precipitation layer, and with a mean mass diameter close to 2 mm (McFarquhar et al., 2007). For this case, we therefore arbitrarily consider an ice crystals fall velocity $v_i = 0.5$ m s$^{-1}$ (Vázquez-Martín et al., 2021).

### 2.2.2   The ISDAC case

The second case is an 8 h long semi-idealized case based on observations from the Indirect and Semi-Direct Aerosol Campaign (ISDAC) (McFarquhar et al., 2011) and focuses on a single layer of stratocumulus cloud deck that formed over pack ice north

of Utqiagvik, Alaska, on 26 April 2008. Clouds were located within a well-mixed layer decoupled from the sea-ice surface. It was set-up in Ovchinnikov et al. (2014) for an intercomparison study of LES with bin or 2-moment bulk microphysics schemes. The initial profiles of temperature, moisture, and horizontal wind components are based on aircraft observations in the mixed layer and radiosoundings at Utqiagvik. The initial moisture profile contains a layer supersaturated with respect to liquid water between $\approx$ 700 and 800 m a.g.l.. Surface turbulent sensible and latent heat fluxes are set to zero and a moderate subsidence of

0.4 cm s$^{-1}$ is prescribed above 800 m a.g.l with a gradual decrease to 0 down to the surface. An important difference in the set-up compared to M-PACE is that the horizontal wind components, temperature and moisture profiles are nudged towards the initial profiles and nudging coefficients are specified to have the height dependency (see Appendix of Ovchinnikov et al., 2014). At the end of the 8 h of simulation, the LES involved in the Ovchinnikov et al. (2014)'s intercomparison exercise exhibit LWP and IWP values ranging roughly between 10 and 50 g m$^{-2}$ and 2 and 20 g m$^{-2}$ respectively, depending on the

ice crystals' number concentration prescribed. Albeit initially built for LES, the almost same set-up is used here to force the LMDZ SCM. We however make two exceptions with respect to the Ovchinnikov et al. (2014)'s protocol. First, we do not prescribe the ice nucleation rate and the ice number concentration as the number of ice crystals is diagnosed in the new cloud phase partitioning scheme. Nonetheless, the INP parameterization used with a default $N_{aero5}$ value of 1 scm$^{-3}$ provides an ice number concentration value at cloud-top temperature around 0.6 L$^{-1}$, which is fairly close to the approximate 1 L$^{-1}$ mean

in-cloud ice crystal number concentrations estimated from multiple measurements during ISDAC (McFarquhar et al., 2011). Second, we activate ice processes from the beginning of the run – and not 2 h later – as ice and liquid phases are treated jointly in the LMDZ cloud scheme. Note that LMDZ ISDAC simulations are run with a 1-min time step and not with the 10-min time step commonly used for the LMDZ SCM simulations. Justification for this choice is provided in Appendix A. Precipitating

ice crystals during ISDAC mostly correspond to unrimed and unaggregated dendrite crystals (Lawson, 2011; Fridlind and Ackerman, 2018) whose size rarely exceeds a few hundred $\mu m$. For this case, we therefore consider an ice crystals fall velocity $v_i = 0.2 \, \mathrm{m \, s^{-1}}$ (Vázquez-Martín et al., 2021), which roughly corresponds to the mean-mass ice crystal velocity of the SAM LES presented in Ovchinnikov et al. (2014).

### 2.2.3 LMDZ SCM simulations and Perturbed Parameters Ensemble experiments

SCM simulations are run with the standard 95-level vertical grid of LMDZ with model layers' thickness increasing from $\approx$10 m near the surface to $\approx$100 m at 2000 m above ground level (Hourdin et al., 2019). Forcings for the MPACE and ISDAC single-column cases have been formatted to the international DEPHY-SCM standard (https://github.com/GdR-DEPHY/DEPHY-SCM/) so that they can be run by any SCM designed to read DEPHY-SCM inputs. Results of SCM simulations – especially LWP and IWP – will be compared to estimates (ranges of plausible values) from observations or reference LES simulations. Nonetheless, the ability of LMDZ to capture the overall structure of MPCs as well as realistic values of LWP and IWP necessarily depends on calibration choices, namely arbitrary choices for tuning parameter values.

This question is even more critical for the new parameterization of cloud phase partitioning that introduces new tuning parameters and that involves direct interactions between the cloud scheme, the shallow convection scheme and the vertical diffusion scheme.

The advantage of numerically-cheap and quick SCM simulations is that the parameteric sensitivity can be explored quite easily using Perturbed Parameter Ensemble (PPE) experiments, by randomly sampling the values of a given set of parameters within pre-determined ranges of plausible values. Then remains the question of the choice of parameters to include in the PPE, namely those for which we should assess the sensitivity. M-PACE and ISDAC boundary-layer clouds structure, water content and phase are expected to mostly depend on the calibration of the turbulent diffusion scheme's parameters, shallow convection scheme's parameters and cloud scheme's parameters including those controlling the subgrid distribution of water vapour (or saturation deficit), the autoconversion to precipitation and its evolution as well as the parameters of the cloud phase parameterization. Exploring with no *a priori* the full ranges of plausible values for the full set of possibly-impacting parameters is however misleading as parts of the explored parameters space, even if leading to reasonable LWP and IWP values for M-PACE and ISDAC clouds, may be climatically not viable – because leading to unrealistic climate states in 3D global simulation – and/or lead to unrealistic structures of other cloud types. To illustrate the sensitivity of the new cloud phase parameterization, we will therefore choose a model configuration in which all parameters of the shallow convection, turbulent diffusion, and cloud scheme that are not specific to ice and mixed-phase processes have been tuned on a series of warm boundary-layer cloud cases as in Hourdin et al. (2021). Only 7 parameters related to cloud phase parameterization and ice precipitation will therefore be retained for the PPE exercise, together with $N_{aero5}$ which controls the number of INPs. Table 1 provides the list of those parameters, their range of acceptable values and their default value. Note that using a toy-model version of the parameterization, Raillard et al. (2025) show that the most critical parameters for the cloud phase partitioning in non shallow-convective boundary-layer MPCs are the parameter $\gamma_{\tau d}$ controlling the Lagrangian turbulent decorrelation time-scale $\tau_d$, the capacitance parameter of ice crystals $C$ as well as the fraction of snowfall $\gamma_s$ that is taken into account in the pre-existing ice

**Table 1.** Name, definition and units, range of acceptable values and default value for the adjustable parameters kept for the PPE exercise.

| Name | Definition | Range | default value |
|---|---|---|---|
| $\gamma_{\mathbf{E}}$ | controls the contribution of plumes' detrainment to supersaturation [-] | $[0.5 - 1.5]$ | 1 |
| $\gamma_{\tau_{\mathbf{d}}}$ | controls the turbulent production of SLW [-] | $[0.15 - 15]$ | 10 |
| $\gamma_{\mathbf{s}}$ | controls the fraction of snowfall for the supersaturation relaxation term [-] | $[0 - 1]$ | 0.1 |
| $\mathbf{C}$ | ice crystals' capacitance parameter [-] | $[0.4 - 1]$ | 0.5 |
| $\mathbf{c_{sub}}$ | controls the magnitude of snowfall sublimation [-] | $[5\ 10^{-5} - 1\ 10^{-3}]$ | $5\ 10^{-4}$ |
| $\mathbf{f_v}$ | controls the ice crystal's autoconversion rate [-] | $[0.5 - 2.]$ | 0.8 |
| $\mathbf{N_{aero5}}$ | controls the number of INPs [scm$^{-3}$] | $[0.1 - 10.]$ | 1.0 |

crystal mass concentration for supersaturation relaxation term. Following Hourdin et al. (2021), the number of simulations of the PPE is 10 times the number of parameters, namely 70.

## 3 Results

Results of LMDZ SCM simulations on the M-PACE and ISDAC cases are now analyzed. Simulations with the default phase partitioning based on the continuous function of temperature will be referred to as CTRL. In the following, simulations using the standard phase partitioning parameterization of Raillard et al. (2025) are referred to as R25, while those using the adapted version, which includes the contribution of plume detrainment to in-cloud supersaturation, are referred to as TEST. Let's recall that the key parameters of LMDZ's physics – excluding parameters restricted to cold cloud physics – have been tuned on boundary-layer and warm cloud cases as in Hourdin et al. (2021). Parameters specific to the new phase partitioning parameterization are set to their default values (see Table 1). Note that TEST and R25 differ only in that $\gamma_E = 0$ in the latter.

### 3.1 M-PACE simulations of surface-coupled Arctic MPCs

Figure 2 shows the vertical profiles of potential temperature, relative humidity, liquid water content (LWC) and ice water content (IWC) during the well-developed phase of the cloud at the end of the M-PACE simulation. Note that radiosonde observations shown in panels a and b by contrast correspond to the initial state of the cloud (17:00 UTC, 9 October 2024) and to the initial potential temperature profile that was used to initialise the simulations. Klein et al. (2009) highlight a strong east-west gradient in cloud base, cloud top and cloud thickness with an overall increase in these quantities along the downstream direction. One can thus expect an overall increase in boundary-layer height during the M-PACE case as cloud forms and deepens, but no reference observational profile of potential temperature corresponding to the end of the case is available.

The CTRL simulation (orange curves) reproduces a cloud of mixed-phase composition, nonetheless with a strong underestimation of the LWC (Figure 2c) and an overestimation of the IWC (Figure 2d). As the cloud formation in the CTRL configuration corresponds to a saturation adjustment process with respect to the ice phase, the model unrealistically simulates a cloud layer close to saturation with respect to ice (Figure 2b). The profile of potential temperature reveals an increase in boundary layer

height with respect to the initial conditions of the simulation that correspond to the observed potential temperature and relative humidity profiles (Figure 2a).

The R25 simulation (blue curves) produces an almost fully-glaciated cloud with an overly high IWC and with a cloud layer near saturation with respect to ice. Only a small patch of LWC is noticeable near cloud base due to the presence of TKE in the lowermost part of the cloud allowing for significant variance of in-cloud supersaturation despite a negative mean value (Figure

3b,c). The boundary layer does not deepen (Figure 2a) in particular due to the absence of cloud-top TKE and entrainment. Additional sensitivity tests to the value of free parameters – in particular $N_{aero5}$ – show that those biases cannot be attributed to calibration issues (not shown). It is worth noting that a pronounced underestimation of the LWC on M-PACE simulations with the Met Office Unified Model was also shown in the study of Furtado et al. (2016) from which the R25 cloud phase partitioning parameterization was inspired.


The TEST simulation exhibits clear improvement with respect to aircraft measurements in terms of LWC and IWC with a clear liquid-dominated upper part of the cloud and ice virga below cloud base (Figure 2c,d). The TEST configuration simulates a deepening of the boundary layer during the run, hence the higher simulated thermal inversion during the well-developed phase of the clouds compared to the observed potential temperature profile (Figure 2a) that again corresponds to the initial

state of the simulation. The TEST simulation also exhibits a cloud layer supersaturated with respect to ice, which qualitatively agrees with observations (Figure 2b). It also exhibits a dryer atmospheric surface layer due to an enhanced upward transport of moisture by shallow convection that coincides with a weaker ice precipitation flux and sublimation below the cloud layer (Figure 2d). Figure 3b shows that adding the effect of thermals's net detrainment (Figure 3a) on the in-cloud supersaturation diagnostics make it possible to simulate substantial supersaturation in clouds (Figure 3b). The subsequent production of SLW

in the upper part of the cloud enhances the cloud-top radiative cooling and indirectly the buoyancy production of TKE. This production enhances the TKE near cloud top (Figure 3c), within the cloud layer and even above through TKE diffusion. Note however the logarithmic x-axis for TKE in (Figure 3c) and therefore the quite sharp decrease of TKE above the cloud. The substantial turbulent activity within the cloud layer and at its top helps increase the boundary-layer depth (Figure 2a) through cloud-top entrainment. It also favours a positive feedback onto the SLW production through the TKE effect on $\sigma_s^2$ (Figure 3c

and Eq. 17).

A peak in IWC is noticeable near cloud base both in observations and in the TEST simulation although with a weaker amplitude (Figure 2d). This peak corresponds to the region where the in-cloud supersaturation (Figure 3b) becomes close to 0 due to weak supersaturation of the detraining updrafts (not shown) thereby limiting the production of SLW.

The sensitivity of the TEST configuration to parameter values' choice is assessed through the investigation of the PPE. Figure

4a depicts the simulated LWP and IWP values at the well-developed phase of the clouds for each of the members of the PPE. It shows that most of members fall in the likely ranges of LWP and IWP from observations and suggests that the results from the new phase-partitioning parameterization with the adaptation to include the effect of supersaturation detrainment are robust. The dependency of LWP and IWP to each of the considered parameters is then investigated (not shown). The analysis reveals that the most determining parameter is $\gamma_{\tau_d}$, particularly for the IWP (Figures 4b and c). The comparison between the R25 and

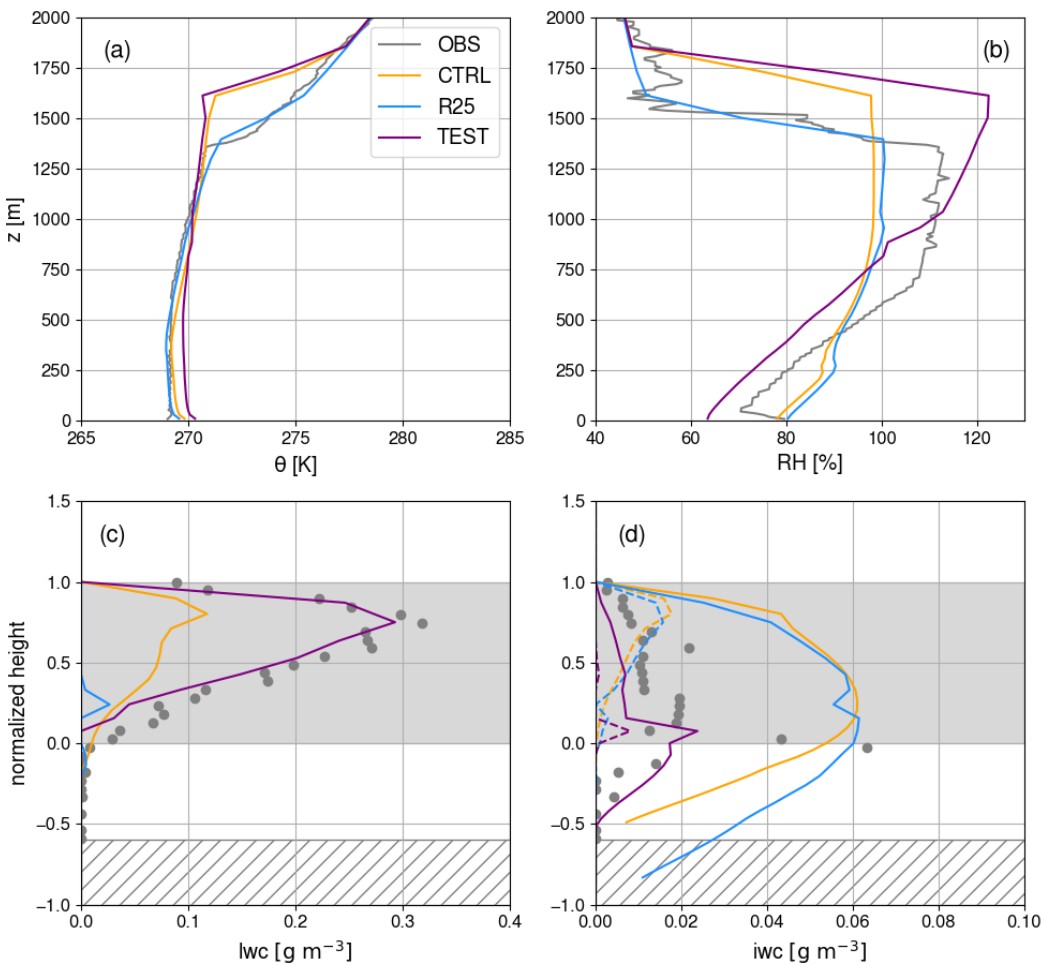

**Figure 2.** M-PACE vertical profiles of potential temperature (a), relative humidity with respect to ice (panel b), liquid water content (panel c) and ice water content including (resp. excluding) snow precipitation in solid (resp. dashed) lines (panel d). CTRL, R25 and TEST simulations are shown in orange, blue and purple respectively and simulation profiles are averaged over the well-developed phase of the clouds, namely between 01:00 and 05:00, 10 October 2024. In panel a and b, observations (in grey) correspond to the radiosonde launched at Utqiagvik, Alaska at 17:00 UTC, 9 October. In panels c and d, the vertical axis used is the normalized height – where −1 is the surface, 0 is cloud base, and 1 is cloud top – for consistency with Klein et al. (2009) and grey shading indicates the location of the cloud. Observations (grey dots) correspond to airborne measurements that do not extend below -0.6 in normalized height, hence the hashed area in panel c and d. Note also that in LMDZ profiles of relative humidity correspond to a diagnostic variable computed as the ratio of the mean specific humidity in the mesh to the saturation specific humidity at the mean temperature in the mesh.

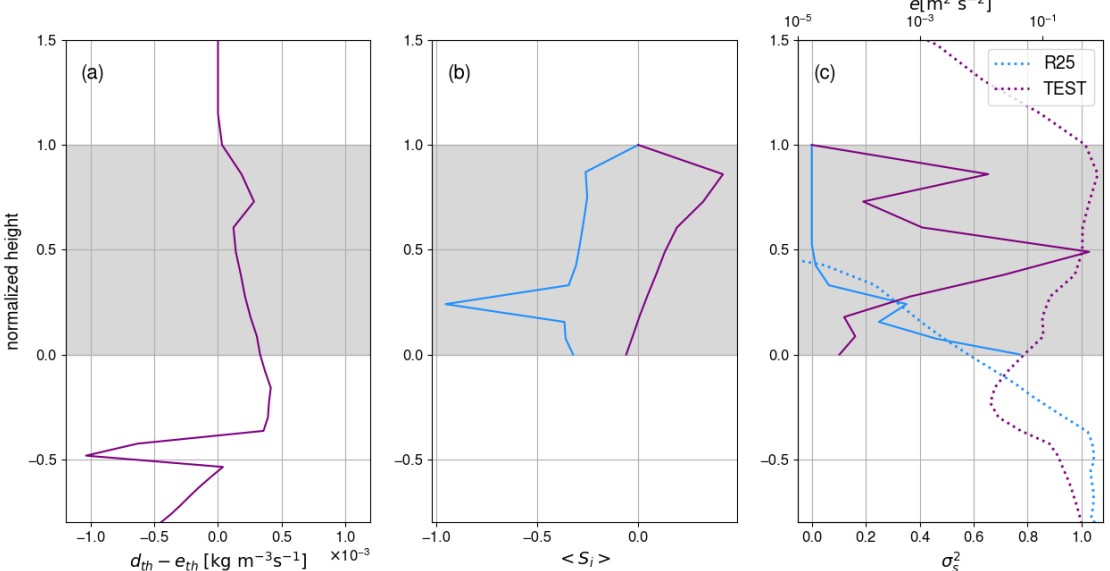

**Figure 3.** (a): M-PACE Vertical profile of net detrainment rate in the TEST simulation (solid line, bottom x-axis). (b): M-PACE vertical profile of the mean in-cloud supersaturation in the R25 (blue) and TEST (purple) simulation. (c): M-PACE vertical profile of the variance of the in-cloud supersaturation (solid lines, bottom x-axis) and of the TKE in the R25 and TEST simulations (dotted lines, top x-axis). In all panels, the vertical axis is the normalized height and grey shading indicates the cloudy area. The profiles shown correspond to the median profiles over the well-developed phase of the clouds, namely between 01:00 and 05:00, 10 October 2024.

TEST simulation shows that it is the inclusion of the contribution to plume detrainment to the in-cloud supersaturation that makes it possible to capture significant SLW content. This might suggest a prevailing dependency on the $\gamma_E$ parameter which controls the contribution of plumes' detrainment to supersaturation. However, the plumes' detrainment enables the triggering of SLW production near cloud top whatever the $\gamma_E$ value within the considered $[0.5 - 1.5]$ sampling interval. The SLW content is then controlled by TKE production (and thus by $\gamma_{\tau_d}$) which determines the vigour of a positive feedback loop, that involves SLW induced cloud-top radiative cooling, enhanced TKE through buoyancy, and increased SLW production driven by subgrid turbulence (TKE).

### 3.2 ISDAC simulations of surface-decoupled Arctic MPCs

We now evaluate the LMDZ SCM simulations on the surface-decoupled ISDAC case. It is worth noting that shallow convection, triggered by surface convective instability is not activated in the ISDAC simulations and therefore, the R25 and TEST simulation give exactly the same results. This section thus mostly aims to assess whether and how the R25 parameterization helps capture the structure of surface-decoupled MPCs. Figure 5 shows the time evolution of the vertical structure of the specific contents of supercooled liquid droplets and ice crystals in the CTRL and TEST simulations. In the CTRL simulation, the formation of

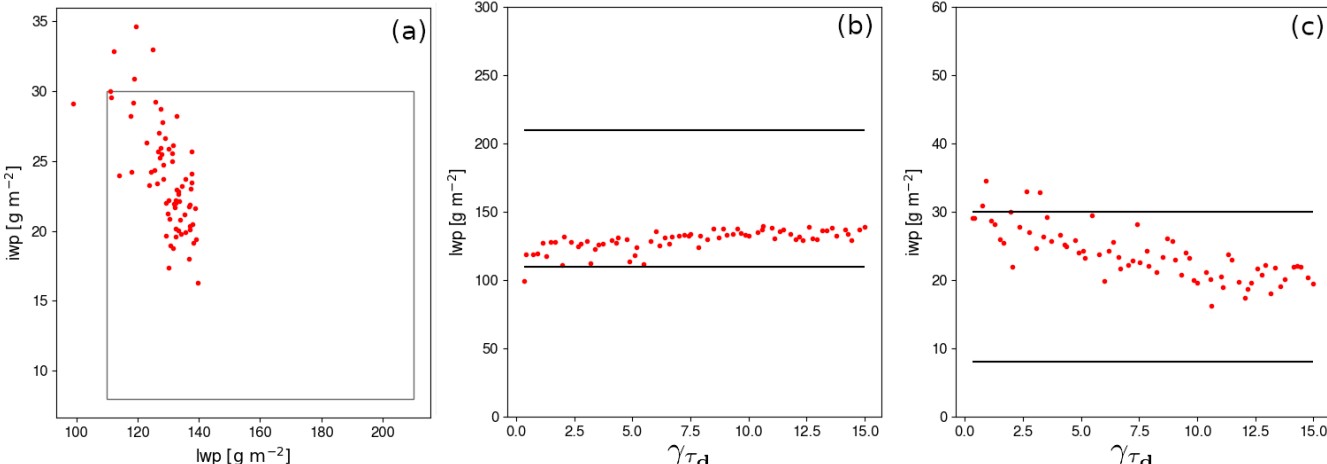

**Figure 4.** (a) Scatter plot of the mean IWP versus LWP during the well-developed phase of the clouds (between 01:00 and 05:00, 10 October 2024). Each red dot corresponds to one amongst the 70 members of the M-PACE PPE. In panel b (resp. c) the LWP (resp. IWP) is plotted versus the value of the $\gamma_{\tau_d}$ parameter (see Eq. 11). The rectangle in panel a and the black horizontal lines in panels b and c indicate the likely range of the regionally averaged LWP and IWP from airborne and ground-based observations according to Klein et al. (2009). Note that the simulated IWP includes the mass of snow precipitation and it considers only the first 2000 m above ground level to avoid accounting for mid- and high-troposphere clouds which can be artefacts as they are not the focus of the case.

clouds with the temperature-based phase-partitioning leads to a relatively dense cloud of mixed phase composition between 550 and 850 m during the first hour of the simulation Figure 5a,b). Cloud formation through saturation adjustment with respect

to ice results in high in-cloud condensed water contents. In turn, this leads to substantial autoconversion of ice crystals into snowfall and of supercooled liquid droplet into supercooled drizzle, which immediately freezes. Moreover, an excessive ice water content near cloud top - whose temperature ranges between 258 and 260 K - is also expected due to the temperature-based phase partitioning that predicts a cloud ice mass fraction of about 30% (Madeleine et al., 2020). As a result, high $q_i$ values and intense ice precipitation are present from cloud top down to the surface. The consequence is that the cloud condensed water

depletes rapidly as there is no water supply. In particular, the absence of significant turbulence below cloud base (not shown) precludes the vertical transport of moisture from low levels up to the cloud. Therefore, the cloud disappears after about one hour of simulation. A shallow ice fog then forms within the stable boundary layer but the overall structure of those clouds does not ressemble the liquid-topped stratocumulus clouds during ISDAC.

Conversely, the TEST simulation captures the formation of a 150-to-200 m deep liquid layer cloud at 800 m with increasing

$q_l$ values (Figure 5c) and supersaturated with respect to ice (Figure 6a), sitting on top of a deepening ice virga layer (Figure 5d), in qualitative agreement with LES simulations (see Figure 2 of Ovchinnikov et al., 2014). The production of ice virga is mostly explained by the freezing of supercooled drizzle following supercooled droplets autoconversion. Figure 6b shows that the SLW production at cloud top, enabled by the phase partitioning scheme of Raillard et al. (2025), leads to intense cloud-top radiative cooling. Subsequently, TKE is locally enhanced through buoyancy production (Figure 6c,d), the latter being parameterized

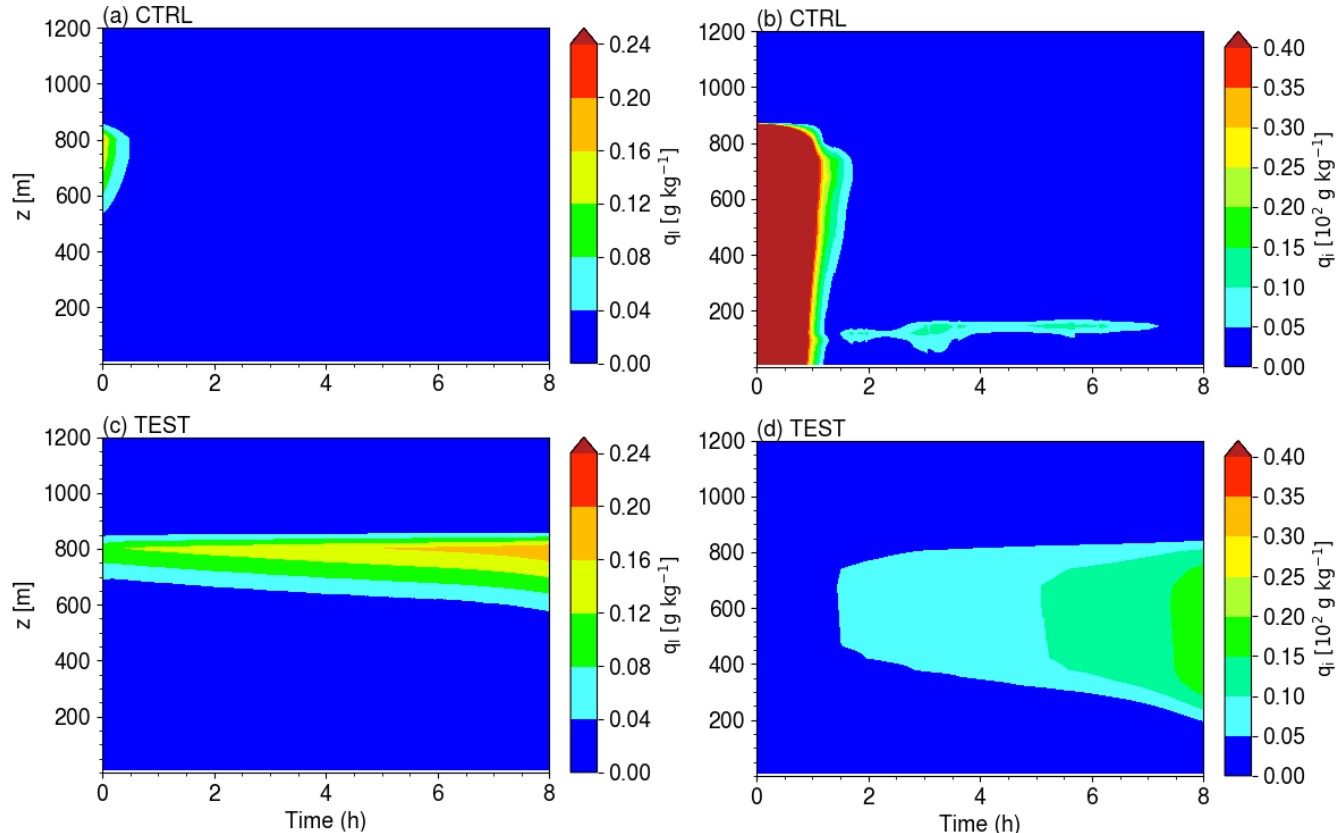

**Figure 5.** Time-height plot of the specific content of cloud liquid water (left panels) and ice water including snow precipitation (right panels) in the CTRL (top) and TEST (bottom) LMDZ SCM simulations on the ISDAC case. Colormap and colorbar's levels are the same as those of Figure 2 in Ovchinnikov et al. (2014) for easy visual comparison with LES results. It is worth recalling that unlike in Ovchinnikov et al. (2014), ice-related processes are allowed from the beginning of the simulations (no 2-h spin-up).

with local K-diffusion formulation (Vignon et al., 2024) which captures only the local component of the cloud-top mixing. This enhanced TKE near-cloud top helps maintain a high variance of in-cloud supersaturation and a liquid ratio close to one. Moreover, the weak but non-negligible top-down vertical transport of TKE by local subgrid turbulent diffusion leads to a net upward turbulent flux of water vapour from the moist lower levels, up to cloud altitude, which favours cloud persistence and deepening. Qualitatively, the TEST simulation thus captures the positive feedback loop involving cloud-top radiative

cooling induced by supercooled liquid droplets, subsequent buoyancy production of turbulence as well as the supercooled liquid water production associated with local turbulence near cloud-top. However, ISDAC LES show that vigorous turbulence is not confined to cloud-top, and that intense turbulent vertical velocity variance extends several hundred meters below the SLW layer (Ovchinnikov et al., 2014). In fact, the mixed-layer forming below the cloud during ISDAC mostly consists in non-local convective cells triggered by radiative cooling at cloud top. In the absence of surface convective instability, LMDZ

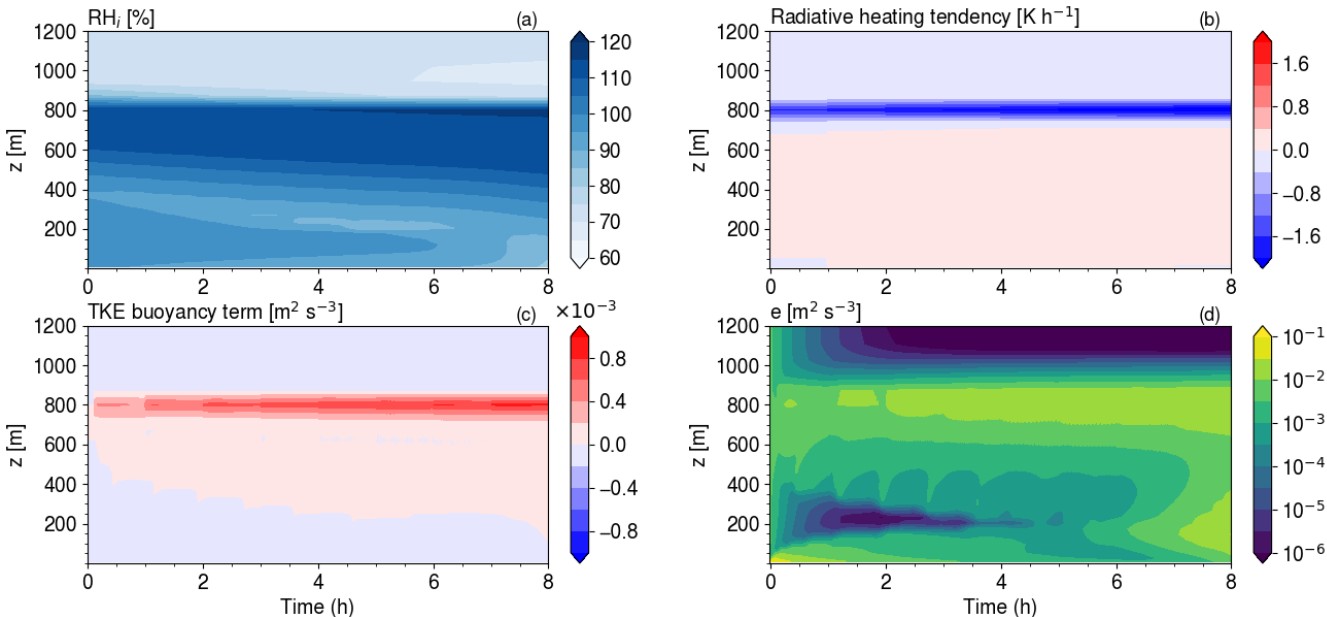

**Figure 6.** Time height plots of the relative humidity with respect to ice (a), radiative heating tendency (b), TKE production term by buoyancy (c) and TKE (d) in the TEST LMDZ SCM simulation on the ISDAC case.

does not account for the contribution of non-local vertical turbulent transport by organized convective cells in addition to the local mixing parameterized by K-diffusion. The non-local component of turbulent mixing is thus missed by our model here.

The LWP and IWP values during the last 2 hours of simulation are 35.7 and 0.93 $\mathrm{g\,m^{-2}}$ respectively. The ISDAC PPE experiment further shows that the LWP and IWP ranges explained by the parametric sensitivity are [35.4 – 35.8] and [0.79 – 0.94] $\mathrm{g\,m^{-2}}$ respectively, with $\gamma_{\tau_d}$ being the most determining parameter as for M-PACE (not shown). The simulated LWP and IWP values are however underestimated compared to the values in the ensemble of LES of Ovchinnikov et al. (2014) which roughly range between 36 and 52 $\mathrm{g\,m^{-2}}$ for the LWP and between 1 and 7 $\mathrm{g\,m^{-2}}$ for the IWP (for their so-called ice1 experiment, corresponding to realistic ice crystal number concentration). Varying $v_i$ from 0.1 to 0.3 $\mathrm{m\,s^{-1}}$ does not dramatically change the results. Further sensitivity tests on tuning parameters not considered in the PPE (and thus not listed in Table 1) were carried out, and a slight increase in IWP was found when decreasing the typical time scale for the autoconversion of liquid droplets that then freeze, but this acts to further reduce the LWP. Given the fact that there is almost no surface precipitation thus no net loss of water towards the surface, the underestimation of the overall condensed water content is therefore necessarily explained by an excess in cloud-top drying, due to an excessive cloud-top entrainment, or to an underestimated water supply up to the condensation level. The absence of a dedicated parameterization for non-local convective mixing from cloud-top is likely responsible for an underestimation of the net upward water flux from the underlying moist layers. Parameterizing the cloud-top driven convection such as the convective downdrafts scheme of Wu et al. (2020) is likely a parameterization development priority to further advance the representation of clouds with an intense cloud-top driven

mixing layer. Such a parameterization may also help reduce the TKE at cloud-top and the time-step dependency of our ISDAC simulations (see Appendix A).

## 4 Summary and conclusions

This study assesses the ability of the single-column version of the LMDZ GCM to capture the structure of polar boundary-layer MPCs. Two modeling cases are considered to address two different types of boundary-layer MPCs: the M-PACE case, consisting of a surface-coupled cloud, and the ISDAC case, consisting of a surface-decoupled cloud.

The CTRL configuration using the default temperature-based cloud-phase partitioning strongly underestimates (resp. overestimates) the LWC (resp. IWC) and misses the in-cloud supersaturation with respect to ice on the M-PACE case. It also fails to capture the overall cloud structure on the ISDAC case.

When considering the new physically-based phase partitioning from Raillard et al. (2025), LMDZ succeeds in representing the surface-decoupled cloud on ISDAC, with a liquid-dominated and supersaturated cloud-top and a gradually deepening ice precipitation below. The positive feedback responsible for this cloud structure, consisting in turbulence enhancement induced by the SLW-driven radiative cooling at cloud-top, further favours the local production of SLW near cloud-top. However, vigorous turbulence remains unrealistically confined close to cloud top and the liquid and ice water contents are underestimated with respect to the LES simulations analyzed in Ovchinnikov et al. (2014). Those shortcomings are likely due to the absence of a parameterisation of non-local convection triggered at cloud-top in LMDZ, leading to an overly weak upward turbulent flux of water vapour below and within the cloud. However, the R25 configuration fails in properly capturing the structure of the surface-coupled cloud on MPACE. Accounting for the effect of thermals' plume detrainment when diagnosing the in-cloud supersaturation in the TEST configuration substantially improves the representation of the LWC and IWC profiles, and the model now succeeds in simulating a supersaturated cloud top. The PPE experiment further shows that the results are robust with respect to the tuning of the cold-cloud related model's parameters, and that the parameter that mostly controls the LWP and IWP is $\gamma_{\tau_d}$ which determines the intensity of SLW production term by subgrid turbulence.

Following this SCM study, future work should now focus on testing and evaluating the representation of polar boundary-layer MPCs in 3D regional simulations with the limited area version of the model to assess the behaviour and performance of the new phase partitioning parameterization when interacting with the atmospheric dynamics on realistic case studies, such as cold air outbreak Arctic MPCs. The new phase-partitioning parameterization will then be considered to be included in an upcoming official version of the LMDZ physics, after a thorough tuning exercise following the methodology of Hourdin et al. (2021) - which may need to add additional metrics, i.e. targets, on the ISDAC and M-PACE cases - followed by a global cloud and radiative flux fields validation such as in Madeleine et al. (2020).

Furthermore, Raillard et al. (2025) and Borella et al. (2025) underline that the ice precipitation treatment of LMDZ is very minimalist, if not too coarse, and future parameterization development work will focus on advancing the ice precipitation scheme in the model with additional microphysical and macrophysical considerations. The remaining limits of LMDZ to capture the vertical structure of the turbulence and the cloud water amounts on the ISDAC case also suggest revisiting the

LMDZ shallow convection scheme to account for downward non-local mixing triggered by convective instability at the top of surface-decoupled clouds.

. *Code availability* The current version of LMDZ is freely available from the project website http://www.lmd.jussieu.fr/~/pub under CeCILL licence. The version used for the specific simulation runs for this paper is the svn release 5727 from June, 27th 2025, which can be downloaded and installed on a Linux computer by running the install_lmdz.sh script available here: http://www.lmd.jussieu.fr/~lmdz/pub:./install_lmdz.
sh.

. *Data availability* All observations from the MPACE campaign are publicly available through the ARM user facility (https://adc.arm.gov/discovery/; last access: 13 May 2022). Forcings for the M-PACE and ISDAC single-column cases are provided under the DEPHY-SCM standard at the following link: https://github.com/GdR-DEPHY/DEPHY-SCM/; last access 21 September 2025.

**Appendix A:  Time-step sensitivity on the ISDAC case**

The SCM simulation of the ISDAC case with LMDZ is particularly sensitive to the time step. The sensitivity comes from the turbulent state that is reached at cloud top during the first hour of the simulation. Figures A1a and b depict the time evolution of the cloud top TKE production terms and TKE in simulations with a 10 min and a 1 min time step with identical tuning parameters' value. In both simulations the cloud-top TKE value after 1 hour results from an equilibrium between buoyancy production ('Buoy' term), TKE dissipation ('Dissip' term) and vertical turbulent transport ('Trans' term) which reflects the
vertical diffusion of TKE by turbulence. However the magnitude of the TKE production terms and of the TKE differ in the two simulations. In fact, once the almost fully-liquid cloud forms at the first time-step, the subsequent cloud-top instability leads to a substantial TKE buoyancy production. In the 1-min time step simulation, this production is balanced in a few minute due to dissipation and – mostly downward – transport term. The 10-min time step is too long to capture this rapid adjustment, enabling the TKE to increase which in turns strengthens turbulent buoyancy flux and the TKE buoyancy production which
overall leads to almost stabilized TKE values after one hour of simulation which are substantially higher.

Regarding the effect on the liquid and ice cloud water contents, changing the time step from 1 to 10 min does not degrade the overall vertical structure of the cloud which keeps a liquid-dominated top and ice precipitation below (Figure A1c,d). However, the simulated $q_l$ and $q_i$ values become weaker (compare panels c and d of Figures A1 and 5), which is particularly due to the increase in cloud-top mixing with overlying dry air associated with the increase in cloud-top TKE (Figure A1a).

. *Author contributions* EV: parameterization adaptation, method, supervision, visualisation, writing. LR: parameterization development, method, analysis, visualisation, writing. AB: LMDZ cloud scheme development, review and editing. GR: original parameterization design, review and editing. JBM: original parameterization design, review and editing.

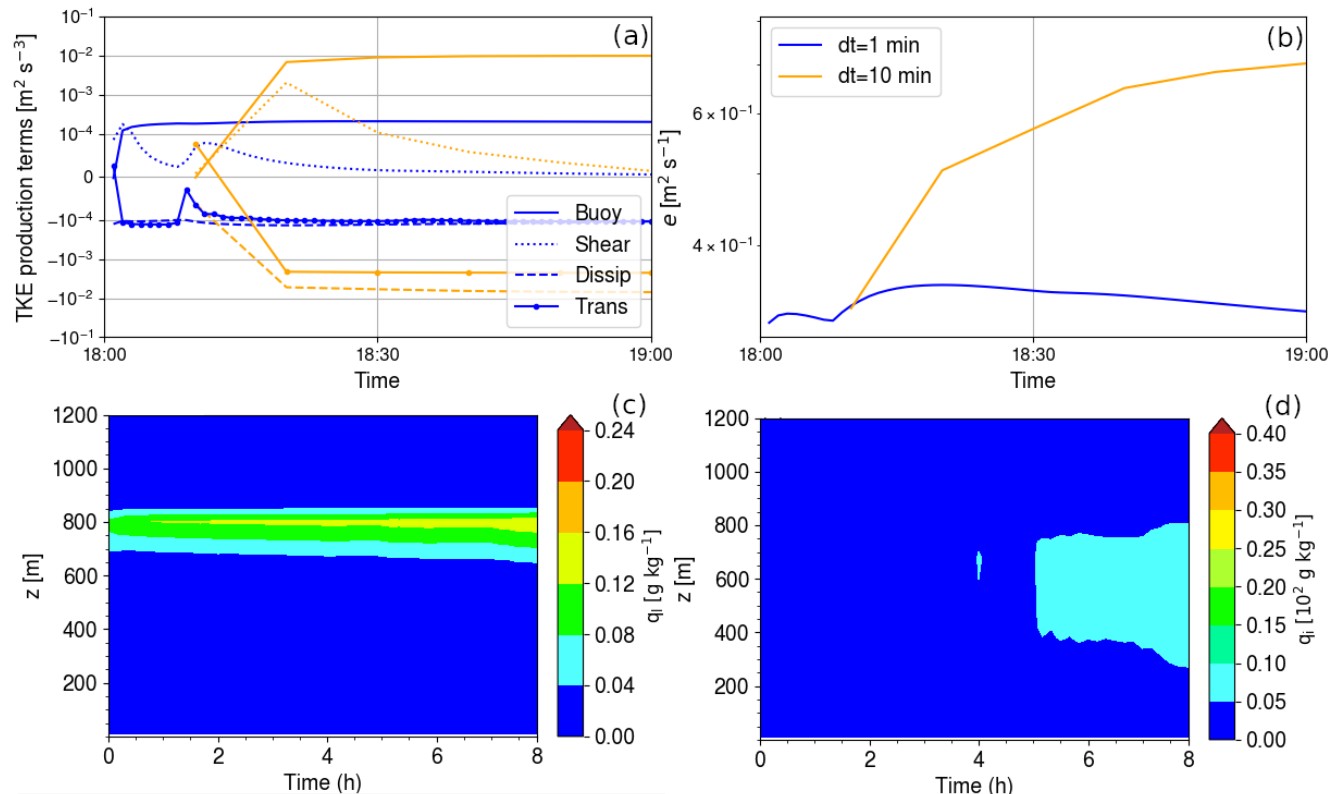

**Figure A1.** Evolution of the TKE production and loss terms (panel a) and of the TKE (panel b) during the first hour of ISDAC simulations at the model level corresponding to cloud top. Yellow (resp. blue) curves correspond to the simulation with a time step of 10 (resp. 1) min. Panels c and d are the same as panels c and d in Figure 5 but for simulations with a time step of 10 min.

*. Competing interests* The authors declare they have no competing interests

.

*. Acknowledgements* This research has been financed by the CNRS-INSU LEFE DEMONIAC project. We acknowledge support from the DEPHY research group, funded by CNRS/INSU and Météo-France, as well as from the PEPR TRACCS project (no. ANR-22-EXTR-0008 funded from the Agence Nationale de la Recherche - France 2030) for supporting the DEPHY-SCM standard. Romain Roehrig is gratefully thanked for helping implement the M-PACE and ISDAC cases into the DEPHY-SCM standard. We thank two anonymous reviewers whose insightful comments helped improve an earlier version of this manuscript.

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
