# Peer review of "Modeling the Coupled and Decoupled states of Polar Boundary-Layer Mixed-Phase Clouds"

_EGUsphere, 2025_

## Author Comment (AC1)

**Revision of**
**'Modeling the Coupled and Decoupled states of Polar Boundary-Layer Mixed-Phase Clouds '**

Etienne Vignon, Lea Raillard al.

December 18, 2025

This document contains the response to the second review of 'Modeling the Coupled and Decoupled states of Polar Boundary-Layer Mixed-Phase Clouds ' submitted to EGUSPHERE for possible publication in Atmospheric Chemistry and Physics. Comments from the Reviewer are in black and answers are in blue. Paragraphs that have been added or modified during the revision process are copied in purple.

**Reviewer #2**

Overview

The authors present a clearly written model development work that relies upon single-column model (SCM) simulations of two well-studied cases of mixed-phase cloud (MPCs). Their modeling approach takes on the revision of phase partitioning to account for the production of supercooled liquid water in updraft regions of turbulent layers that may or may not be coupled to the surface. Another approach that has also proven effective (e.g., Silber et al. 2022 appendix) is application of microphysical process rates in a procedure where a moist turbulence scheme operates on thermodynamic and microphysics fields (e.g., mixing ice species), cloud liquid water is diagnosed from a macrophysics scheme (rapid equilibration), ice formation rate is diagnosed from ambient aerosol-modulated immersion freezing (in addition to multiplication schemes), ice cover is diagnosed by macrophysics, and ice growth then follows from thermodynamic conditions with associated sedimentation offset by turbulent mixing. In the latter approach, phase partitioning is only indirectly affected by turbulence. It would be very interesting to see this new parameterization approach tested against other approaches in a case with rapidly evolving boundary layer depth and cloud-top temperature, such as the ongoing COMBLE-MIP community project (https://arm-development.github.io/comble-mip/README.html). I see no major methodological flaws in the presented work and recommend publication of this relevant work after addressing my comments below (especially comment 3).

We would like to sincerely thank the referee for the review of our manuscript and insightful comments. Please find here below or responses to the comments. We completely agree that assessing the behaviour of the new parameterization in a rapidly evolving boundary layer would be very interesting. In fact this is a work we are planning to do considering two case studies of warm air intrusion and cold conveyor belt during the HALO-AC3 campaign https://halo-ac3.de/halo-ac3/campaign/. Even though this is beyond the scope of the present manuscript, we followed your recommendation and included the COMBLE cold-air outbreak case study to our library of SCM cases. We are able to run LMDZ on this case, and assess how the new parameterization captures the structure of the convective MPC. The left panel in Figure 1 below shows that our model captures the formation of a mixed-phase cloud layer along the cold air outbreak trajectory – note that it is a Lagrangian case – that gradually deepens. Ice precipitation increases in the second part of the simulation, along with a decrease in cloud fraction (not shown) which qualitatively agree with the transition from rolls to open-cell clouds during the case. The ice water path and ice precipitation flux (right panels in Figure 1) have a magnitude comparable to that of LES (see https://arm-development.github.io/comble-mip/README.html). However, to properly evaluate the model and set LMDZ in a broader international framework of MPC modelling, we would like to officially participate in the intercomparison exercise and benefit from the jupyter-notebook evaluation tools developed by the COMBLE community. This is a request we are planning to send very soon.

[Figure]

Figure 1: Illustration of recent LMDZ SCM simulations on the COMBLE cold-air outbreak case. The left panel shows a time height plot of the specific liquid water content (shading) and of the ice precipitation flux (contours). The right panels show the time series of the liquid water path and surface snowfall.

1. Regarding the second paragraph of the introduction, I would urge the authors to consider another much simpler approach to explaining polar mixed-phase clouds (e.g., Silber et al., 2021), namely by considering their features as a readily understood outcome of weak heterogeneous ice formation from their familiar warm (liquid-only) counterparts. This allows a more intuitive explanation of many widely observed key features. For instance, in a well-mixed coupled case in the limit of weak ice formation (approaching the warm cloud case), cloud liquid is roughly adiabatic (consistent with meteorology 101). Once heterogeneous ice formation is sustained in the immersion mode at the very weak levels it is typically observed (cf. Silber et al., 2021), it serves as a very weak sink of moisture and naturally "the liquid remains at the top". This also readily explains the resilience, which I think should not be unexpected at all and is reproduced by higher resolution models. I therefore suggest to avoid furthering the "unexpected" idea by repeating it here, because it is based on the mistaken baseline assumption that there is no relevant time scale to ice growth and sedimentation. Taking the warm case as a foundation also readily explains the commonality of continuous liquid bases observed by lidar (see examples in Silber et al., 2021), which becomes circuitous in the current explanation owing to dependence on updrafts (note: adiabatic liquid water content in a well-mixed layer is independent of updraft strength). In my opinion, referring to WBF also introduces an unnecessary overlay that is not encoded in models as a "process" because models don't need to add anything to the basic physics: namely, ice is growing everywhere that relative humidity exceeds saturation with respect to ice, continuously both above and below any supercooled liquid cloud bases and there is no need to consider any further explanation separately within versus below liquid cloud base.

Thank you for this insightful comment. We fully agree with your point. The fact that, in the limit of weak ice formation, cloud liquid water remains roughly adiabatic explains why models with a well-tuned shallow convection scheme are able to reasonably reproduce the cloud liquid water content and structure—provided that unrealistic ice partitioning schemes, such as purely temperature-based ones, are deactivated or replaced with more physically based microphysics. In addition, this also explains the rather weak parametric sensitivity of the simulated LWP in the surface-coupled and liquid-dominated M-PACE case in our LMDZ simulations (see Fig. 4a). Similarly, we agree that the WBF process is not directly parameterized in models, but rather emerges from the parameterized water deposition as soon as the relative humidity with respect to ice exceeds 100%, which occurs near supercooled liquid layers. Nonetheless, we think that the originally 'surprising' aspect of the resilience of supercooled droplets in clouds mostly stems from the scarcity of ice nucleating particles and of the frequently weak role of heteogeneous freezing processes. And in fact, the liquid content becomes dependent on updrafts strength as soon as the ice crystal content becomes significant [**Korolev˙2008**] but this does not concern the liquid-dominated MPCs found in many polar boundary-layer contexts. However, this effect becomes non negligible in ice-dominated mid-level or deep clouds such as nimbostratus clouds (e.g., [**Gehring˙2020**]). Following your recommendation,

the paragraph in the Introduction has been rephrased as follows:
Observational campaigns at the poles have revealed the resilience of boundary-layer MPCs which can persist for several days. This resilience could *a priori* be surprising given the thermodynamical instability of SLW droplets at $T < 0°C$ and their depletion through vapor deposition towards ice crystals as the relative humidity with respect to ice exceeds 100 % [**Shupe·2006**, **Morrison·2012**]. The formation of SLW in polar boundary-layer clouds results from interactions between turbulence, microphysics and radiation [**Korolev·2017**]. In turbulent updrafts, generated either by convective instability at the surface [**Shupe·2008**] or by cloud-top eddies induced by radiative cooling [**Simpfendoerfer·2019**, **Barrett·2020**], the relative humidity can reach saturation with respect to liquid through air adiabatic cooling during ascent [**Korolev·2003**]. Cloud droplets can thus form almost adiabatically and are advected upward, thereby forming a thin – a few hundred meters deep – liquid layer. Most of the time, the scarcity of ice nucleating particles (INPs) [**Eirund·2019**, **Creaman·2022**, **Wex·2025**] in polar regions makes heterogeneous freezing processes weakly active. Subsequently, the vapor deposition overall serves as a very weak sink of moisture [**Silber·2021**] which explains the commonality and resilience of liquid-bearing clouds [**Silber·2020**] at the poles. The growth of ice crystals through vapour deposition and riming [**Maherndl·2024**, **Chellini·2024**] make them sediment below – and separate from – the liquid layer.

2. Regarding the introduction to model capabilities (lines 60), I would suggest to add more than one reference showing that many higher resolution models can perform very well indeed for both coupled and decoupled MPCs as long as ice formation rate and ice properties are realistic. For instance, for a coupled case, Tornow et al. (2025) illustrate mixed-phase simulations that can reproduce basic features of sustained liquid water path, precipitation onset and subsequent cloud cover and droplet number concentration reduction. For a decoupled case, Silber et al. (2019, 2020) present large-eddy simulations that reproduce the progressive development of supercooled liquid in a stable layer, turbulence onset, and in that case, the sustained coexistence of liquid and ice precipitation processes. I would also add that the Lagrangian approach taken in those studies provides a strengthened foundation for large-scale model development because it allows a test of whether a mixed-phase cloud can realistically form within an initially cloud-free environment and proceed to reproduce observed transitions. Silber et al. (2022) also illustrate that the NASA ModelE3 GCM code can well reproduce the decoupled cloud case in single-column model (SCM) mode (see appendix), including onset of turbulence and co-existing precipitation in two phases.
Thank you for this suggestion. Following your recommendations, the corresponding paragraph in the Introduction has been modified as follows:
Some large eddy simulation (LES) models, cloud-resolving models and mesoscale models can reasonably capture the structure of both surface coupled and surface-decoupled boundary-layer MPCs as long as ice formation rate and ice properties are realistic (e.g., [**Klein·2009**, **Ovchinikov·2014**, **Arteaga·2024**, **Silber·2019·2 Tornow·2025**]).

However, General Circulation Models (GCMs) still struggle to simulate the vertical structure and microphysical properties of surface-coupled clouds (e.g., [**Liu˙2011**, **Gettelman˙2015**, **Zhang˙2020**]). These shortcomings in GCMs are even more pronounced for surface-decoupled MPCs, even though recent single-column simulations with the NASA ModelE3 model show promising results, including onset of turbulence from a purely liquid stratiform cloud and subsequent triggering of ice precipitation [**Silber˙2022**]. Overall, shortcomings in the representation of polar boundary-layer MPCs in GCMs lead to substantial biases in the representation of the surface-based temperature inversion over the wintertime Arctic sea ice in cloudy conditions [**Pithan˙2014**].

3. Regarding the leading problems in large-scale model parameterization, I would have placed first the extreme uncertainty in parameterization of ice formation rate in the immersion mode (e.g., Knopf et al., 2023) and via ice multiplication where it far outpaces the immersion mode (e.g., Korolev et al., 2024; likely same process as in deeper convection from long evidence of colocation with drizzle in MPCs). These are but a few examples of decades of evidence that we cannot capture ice formation rates to one or even several orders of magnitude. No degree of improving other processes can readily cover for that. The Knopf et al. (2023) study further shows how a diagnostic scheme following DeMott type INP parameterizations can produce extremely unrealistic rates of ice formation. Can the authors show that their ice formation rates in these cases are consistent with simple rough estimates of INP source strength and activation rate?

Thank you for this suggestion that we took into account. We now place first the treatment of ice microphysical processes as follows:

Overall, the parameterization of MPCs in GCMs remains extremely challenging and the difficulty mostly lies in: (i) the parameterization of ice microphysical processes [**Forbes˙2014**, **Barrett˙2017**, **Vignon˙2021**], in particular the parameterizations of ice formation rate in the immersion mode (e.g., [**Knopf˙2023**]) and of secondary ice production processes (e.g., [**Pasquier˙2022**, **Sotiropoulou˙2020**, **Possner˙2024**]); Regarding the second part of your comment, we are indeed aware of the limits of the DeMott type INP parameterizations. Note however that in our approach, this parameterization only aims to provide an estimate (diagnostics) of the ice crystal number concentration. We thus cannot directly verify the ice formation rates but we can check whether the prediction of the ice crystal number concentrations concurs with the values measured during the two cases. For ISDAC, we mentioned in the text that the predicted concentration corresponds reasonably well with that measured in **McFarquhar˙2011**. For M-PACE however, the information was missing in the paper. The observed ice crystals number concentration throughout the cloud layer – whose temperature roughly varies from -15 to -10 $^{o}$C – ranges between $10^{-1}$ and 10 L$^{-1}$ with most values of around 1 L$^{-1}$ [**McFarquhar˙2007**]. Our ice crystal estimates in this temperature range using the DeMott parameterization ranges between 0.1 and 2 L$^{-1}$, depending on the prescribed background aerosols concentration (in the perturbed parameter ensemble experiment). Of course, substantial work remains to be done to properly parameterize ice nucleation in our cloud scheme, but we can at least ensure that the predicted ice crystal number concentration is reasonable. We have added the following paragraph in Sect. 2.2.1:

The observed ice crystals number concentration throughout the cloud layer ranges between 0.1 and $10\,\mathrm{L}^{-1}$ with most values of around $1\,\mathrm{L}^{-1}$ [**McFarquhar˙2007**]. The INP parameterization used in our cloud scheme with a default value of 1 $\mathrm{scm}^{-3}$ provides an ice number concentration roughly between 0.1 and 0.6 $\mathrm{L}^{-1}$ for the temperature range within the M-PACE cloud layer, namely between $\approx$ -15 and -10°C. Those values are thus realistic but in the lower part of the measured range. Simulations with higher ice crystals number concentration will be investigated by varying $N_{aero5}$.

4. Regarding the case studies selected (section 2.2), a major difference is that ISDAC included continuous nudging of temperature and water vapor whereas M-PACE applied fixed large-scale advective flux divergence profiles. When applying nudging to LES and SCM at every time step, the model thermodynamic profiles cannot diverge as much from one another; in other words, if divergence grows more in one model, it is more offset. Please clarify for readers in the text whether the authors apply nudging in the ISDAC case and whether that differs from the M-PACE case.

We agree this is an important difference that we should clarify in the manuscript. For M-PACE, we now specify:

The advective forcings of the SCM are prescribed and consist in vertical profiles of temperature and humidity horizontal advection terms as well as that of vertical velocity that are constant in time

and for ISDAC:

An important difference in the set-up compared to M-PACE is that the horizontal wind components, temperature and moisture profiles are nudged towards the initial profiles and nudging coefficients are specified to have the height dependency (see Appendix of [**Ovchinnikov˙2014**])

---

## Author Comment (AC2)

**Revision of**
**'Modeling the Coupled and Decoupled states of Polar Boundary-Layer Mixed-Phase Clouds '**

Etienne Vignon, Lea Raillard al.

December 18, 2025

This document contains the response to the first review of 'Modeling the Coupled and Decoupled states of Polar Boundary-Layer Mixed-Phase Clouds' submitted to EGUSPHERE for possible publication in Atmospheric Chemistry and Physics. Comments from the Reviewer are in black and responses are in blue. Paragraphs that have been added or modified during the revision process are copied in purple.

**Reviewer #1**

This paper evaluates two new microphysical parameterizations in simulations of well-tested M-PACE and ISDAC mixed-phase stratocumulus cases in the LMDZ (global atmospheric component of the IPSL-CM Earth System Model) single column model. In this model, boundary layer turbulent fluxes are parameterized with an Eddy Diffusivity-Mass Flux scheme, where the mass-flux scheme is only active when surface convective instability occurs. Therefore, turbulence in decoupled cloud cases (i.e., the ISDAC case) is only parameterized with local counter-gradient diffusion.

In the current version of the model, phase-partitioning in boundary layer clouds is a function of temperature. A parameterization developed in Raillard et al. (2025) for mid-level clouds that replaces a temperature dependent formulation for one that is a function of subgrid turbulent activity and ice crystal properties is added to the convective boundary layer scheme. The second new parameterization adds a "homogenization" term to the equation for the evolution of supersaturation of ice. This parameterization accounts for air parcels mixing between clouds in the environment and air in the surface-forced thermal plumes. This parameterization was included in Furtado et al. (2016) but not in Raillard et al. (2025). This second parameterization is only active when surface convective instability occurs. Simulations without these new parameterizations is referred to as CNTL. Simulations with the new phase-partitioning scheme is referred to as R25. Simulations with both new parameterizations is referred to as TEST.

Perturbed parameters ensemble experiments are performed for the two case studies to test the sensitivity to parameters that control turbulence and ice crystal properties within acceptable ranges.

We sincerely thank the Reviewer for the thorough and insightful review of our manuscript. We truly appreciated all the comments, which have substantially helped us improve the study. Please find below our detailed responses to each comment.

**Comments**

For the M-PACE case, only the TEST simulation can produce a mixed-phase stratocumulus with cloud liquid and ice similar to the observations. Even though the R25 simulation has a more realistic potential temperature profile, it produces almost no liquid and too much cloud ice. The fixed Naero5 for the M-PACE case should be much lower than for the ISDAC case. 0.16/L is the value typically used for these cases studies. How does R25 perform with lower values of INP? Thank you for raising this point. In fact, changing the INP concentration to 0.16/L does not substantially change the results except that slightly higher ice precipitation occurs due to the overall lower ice number concentration (see figure below). Additional sensitivity tests exploring a wider range of INP values and varying the value of other tuning parameters does not change the overall conclusion (not shown). In fact, neglecting the supersaturation source from the plumes' detrainment (i.e. shifting from R25 to TEST) is paramount to capture the liquid-dominated cloud structure and more reasonable ice concentration. Note that the sensitivity to the INP concentration is thoroughly assessed through the inclusion of the $N_{aero5}$ parameter in the PPE exercise. We have added the following paragraph in Sect. 3.1:
Additional sensitivity tests to the value of free parameters – in particular $N_{aero5}$ – show that those biases cannot be attributed to calibration issues (not shown).

TEST produces relative humidity with respect to liquid in the liquid layer that is greater than 100%. How is this possible?
First, we would like to emphasize that the relative humidity is a diagnostic variable of the model. By construction, the cloud scheme condenses all the water in excess to saturation with respect to liquid. What is happening in the plot is a very subtle issue related to output variables only. In presence of shallow convection, the cloud scheme computes a subgrid distribution of the saturation deficit $s$, and condenses all the water corresponding to $s$ values higher than the $s_{liq}$ threshold, which corresponds to liquid saturation. By construction, this adjustment method makes the saturation deficit never exceed 0 and the relative humidity with respect to liquid never exceed 1 (as $s = q_{sl}(\text{RH} - 1)$). This has been double-checked with systematic prints at the end of the condensation routine (not shown). However, the quantity that is shown in the plot in the paper is not strictly speaking the mean relative humidity in the mesh $\overline{\text{RH}_l}$, but

[Figure]

Figure 1: Same as Figure 2 in the main manuscript but the dark blue curve corresponds to a R25 simulation with a INP concentration of 0.16/L.

a diagnostic variable computed at the end of the parameterizations' sequence that can differ by a few % from $\overline{\mathrm{RH_l}}$: $[\mathrm{RH_l}] = \overline{q}/q_{sl}(\overline{T})$. Here, $\overline{q}$ and $\overline{T}$ refer to the mean specific humidity and temperature in the mesh (which are state variables of the model), and $q_{sl}$ is the saturation specific humidity with respect to the liquid phase. In order to avoid any confusion, the vertical profiles of RHl have been removed from the figure and specifications on how the RHi variable is calculated in the model are now given in the caption.

TEST has near surface layer RH that is much lower than obs and the other runs, indicating too much mixing of sub-cloud air into the liquid layer? This is an interesting point indeed. The investigation of humidity tendencies at the first model level indeed reveals that the shallow convection scheme is more efficient in transporting water upward in the TEST simulation than in

the two other simulations. The exact reason behind this is not completely clear but the weaker ice precipitation flux – that tends to overall dry the system out and which has an associated stabilisation effect due to the sublimation below the cloud – is at least one part of the explanation. Note however that the observed RH profile corresponds to the initial profile of the simulation (see figure below), and that the boundary layer deepens during the run. We do not have any observational reference for the RH vertical profile near the end of the run and thus it is difficult to assess whether this near-surface drying is realistic or not. We have added the following paragraph in the manuscript to clarify this aspect in addition to additional clarification on the difference in timing between the profiles shown in the simulation and that from the observations:

It also exhibits a dryer atmospheric surface layer due to an enhanced upward transport of moisture by shallow convection that coincides with a weaker ice precipitation flux and sublimation below the cloud layer (Figure 2d).

[Figure]

Figure 2: Vertical profile of simulated relative humidity with respect to ice at the beginning of the M-PACE run (black) and 12 hours later (red). Note that the black profile is very close to the radiosounding that served to initialize the profiles in the simulation.

M-PACE has large surface fluxes (cold air outbreak) but is there no representation of cloud-driven top-down turbulence in the model? How does this change the balance between turbulence and microphysics at cloud top in the model?

Although non-local mixing through large eddies — that is, eddies whose size exceeds the typical thickness of model layers — is not taken into account in LMDZ (we elaborate on this aspect in more detail in subsequent responses), local cloud-top turbulence is parameterized through eddy diffusivity using a TKE-l scheme. The combination of a mass-flux representation of boundary-layer convective structures with an eddy-diffusivity scheme — the so-called thermal plume model — has proven successful in representing the structure of stratocumulus clouds in LMDZ, and in particular, the cloud-top dynamics [**Hourdin˙2019**]. **Wu˙2020** further showed that adding a parameterization of non-local downdrafts only marginally improves the overall simulation of surface-coupled warm stratocumulus clouds. Overall, we are therefore confident in LMDZ's ability to represent turbulent transport in the surface-coupled stratocumulus case during M-PACE, and that the absence of parameterized downdrafts is not detrimental to capturing the cloud-top dynamics (again, for surface-based MPCs). Consequently, the main conclusions regarding the interactions between microphysics and turbulence remain robust: the TKE-l diffusion scheme in LMDZ captures the local generation of turbulence at cloud top associated with local convective instability, and supercooled liquid water (SLW) is then produced through the new phase-partitioning scheme, which explicitly relates TKE to SLW production. We have added a paragraph in Sect. 2.1.1 to emphasize the ability of the thermal plume model to capture the dynamics of stratocumulus clouds:

The combination of the LMDZ eddy-diffusivity and mass-flux schemes has proven successful in representing the structure of stratocumulus clouds, and in particular, the cloud-top dynamics [**Hourdin˙2019**].

TEST produces large TKE above the liquid layer in the inversion (Figure 3c). How is TKE calculated?

This is a good point indeed. TKE is calculated based on a typical prognostic equation (see details in **Vignon˙2024**). The mean vertical profiles of TKE production and loss terms during the same period as that shown in Figure 3c are plotted below. In the TEST simulation, the fact that supercooled liquid water is captured near cloud top strongly enhances the cloud-top radiative cooling. Subsequently, convective instability develops near cloud top and TKE is generated through buoyancy production and is transported – by diffusion – above the liquid layer. Note however the logarithmic x-scale for TKE in Figure 3c. In fact despite the diffusion, the TKE strongly decreases above the inversion. The corresponding paragraph in Section 3.1 has been rephrased as follows:

The subsequent production of SLW in the upper part of the cloud enhances the cloud-top radiative cooling and indirectly the buoyancy production of TKE. This production enhances the TKE near cloud top (Figure 3c), within the cloud layer and even above through TKE diffusion. Note however the logarithmic x-axis for TKE in Figure 3c and therefore the quite sharp decrease of TKE above the cloud.

Lines 341-345: How is turbulence due to cloud-top radiative cooling represented in the model? By local diffusion? Or is it in the microphysical scheme,

[Figure]

Figure 3: Mean vertical profiles of TKE production and loss terms on the MPACE in the TEST simulation (the same period as that of Figure 3c in the main manuscript is considered).

determined by the Lagrangian turbulent decorrelation time-scale? Is it assumed that local diffusion estimated from TKE is a representation of non-local mixing by cloud-top cooling? Is so, why is TKE limited to cloud top in the ISDAC simulation?

Turbulence due to cloud-top radiative cooling is represented by local turbulent diffusion, that is by the local TKE-l turbulence scheme, the non-local component being missing (see further explanation in our answer to your last comment on the ISDAC simulations). In this scheme, the TKE production terms – in the TKE evolution equation – are closed with local closure formulations. Similarly to traditional TKE-l schemes, the buoyancy production term is expressed with a K-gradient approach (see Vignon et al. 2024). Therefore, the simulated TKE buoyancy production is located near cloud-top and coincides with the occurrence of strong temperature gradients. In the TKE-l diffusion scheme, the vertical transport of TKE is only ensured by local diffusion, which is a relatively slow process, hence the TKE confined near cloud top in our ISDAC simulations. In the microphysical scheme, the SLW production by local turbulence (2nd term in Eq. 15) is related to the TKE calculated in the TKE-l scheme and is therefore a local production. The text of Sect. 3.2 in the main manuscript has been modified to clarify how the turbulence due to cloud-top radiative cooling is represented in the model:
Subsequently, TKE is locally enhanced through buoyancy production (Figure 6c,d), the latter being parameterized with local K-diffusion formulation [**Vignon˙2024**] which captures only the local component of the cloud-top mixing.

For the ISDAC case, since this is a decoupled case where the surface convective scheme is inactive, R25 and TEST are the same. This simulation tests the impact of the two different phase-partitioning schemes. It is no surprise that the temperature dependent phase-partitioning scheme produces too much cloud ice and causes the liquid layer to collapse.
We agree that it is not that surprising that the temperature dependent phase partitioning produces too much cloud ice which results in an overall too short cloud lifetime. Nonetheless, this phase partitioning is that used in the CMIP6 version of LMDZ. Moreover, many climate models still use a simple temperature function to determine the cloud phase. We therefore deem important to show – even briefly – the effect of such phase partitioning on a simple case of decoupled Arctic boundary-layer cloud. Note that not only the phase-partitioning methodology but also the fact that the cloud water content is estimated through saturation adjustment with respect to the ice phase explain the biases in the CTRL simulations. To emphasize that this result is kind of expected, we have reformulated the corresponding paragraph as follows:
Cloud formation through saturation adjustment with respect to ice results in high in-cloud condensed water contents. In turn, this leads to substantial autoconversion of ice crystals into snowfall and of supercooled liquid droplet into supercooled drizzle, which immediately freezes. Moreover, an excessive ice water content near cloud top - whose temperature ranges between 258 and 260 K - is also expected due to the temperature-based phase partitioning that predicts a cloud ice mass fraction of about 30% [**Madeleine˙2020**]. As a result, high $q_i$ values and intense ice precipitation are present from cloud top down to the surface.

Lines 367-369: Again top-down vertical diffusion of TKE by subgrid turbulence is discussed but isn't this just local diffusion in the model?
Yes it is 'just' local diffusion (see our answer to your first comment on the ISDAC case and the more general discussion in response to your general comment below). This is now specified in the main text:
Moreover, the top-down vertical transport of TKE by local subgrid turbulent diffusion [**Vignon˙2024**] leads to a net upward turbulent flux of water vapour from the moist lower levels, up to cloud altitude, which favours cloud persistence and deepening.

Figure 6c shows TKE buoyancy term and TKE essentially only at cloud top but Ovchinnikov et al. (2014) shows maximum TKE near the liquid cloud base. Even though the R25/TEST simulations maintain a liquid layer, this result indicates fundamental error in turbulence that will also produce fundamental

errors in microphysics.
We address this comment jointly with the last one. Please see our response below.

Minor comments:

Line 98: Change "ofrid" to "of".
Corrected.

Figure 2 and 3: Include "M-PACE" in figure caption.
Added.

Line 343: ". . . loop, that. . . "
Corrected.

I have major questions about the parameterizations used in this climate model. Is it assumed that local diffusion estimated from TKE is a representation of non-local mixing by cloud-top cooling? If so, why is TKE limited to cloud top in the ISDAC simulation? Also, There are many ways to modify turbulence/microphysics in order to maintain cloud liquid. My concern is that this simulation is getting the right answer for the wrong reason. This is extremely important for climate simulations since it will produce unrealistic sensitivity to changes in environmental conditions, surface conditions, and aerosols. Thank you for this general comment that clearly raises the need for additional information in the paper. First, we would like to emphasize that we totally acknowledge that our model misses the non-local component of the cloud-top driven mixing. As explained above, TKE is limited to cloud-top as the buoyancy-production term is expressed with a local closure and because the model does not have a subgrid mass-flux – i.e. convective – parameterization that represents the non-local transport by cloud-top driven convective cells. As explained in the paper, the shallow-convection scheme only activates when convective instability occurs at the surface. To our knowledge, only a few large-scale atmospheric models account for an explicit parameterization of convective downdrafts triggered at cloud top (e.g., **Wu˙2020**). This is an important shortcoming and limitation of our study that should be stated more explicitly at different places of the manuscript and that should appear more clearly as a development priority. However, we do believe that our new parameterization captures the production of SLW for the good reasons, at least qualitatively. Even if it occurs only locally near cloud top, the positive feedback loop involving cloud-top radiative cooling induced by supercooled liquid droplets, subsequent buoyancy production of turbulence as well as the supercooled liquid water production associated with local turbulence is reproduced. Of course, the next step will be to capture this feedback loop with more realistic vertical structure of turbulence considering the non-local mixing by non-local convection triggered at cloud top. This is definitely

in the abstract:

'most of the turbulence is confined near the cloud-top which is probably due to a missing parameterization for convective downdrafts in the model.'

In section 3.1.2:

'Qualitatively, the TEST simulation thus captures the positive feedback loop involving cloud-top radiative cooling induced by supercooled liquid droplets, subsequent buoyancy production of turbulence as well as the supercooled liquid water production associated with local turbulence near cloud-top. However, IS-DAC LES show that vigorous turbulence is not confined to cloud-top, and that intense turbulent vertical velocity variance extends several hundred meters below the SLW layer **Ovchinnikov˙2014**. In fact, the mixed-layer forming below the cloud during ISDAC mostly consists in non-local convective cells triggered by radiative cooling at cloud top. In the absence of surface convective instability, LMDZ does not account for the contribution of non-local vertical turbulent transport by organized convective cells in addition to the local mixing parameterized by K-diffusion. The non-local component of turbulent mixing is thus missed by our model here.'

and:

'The absence of a dedicated parameterization for non-local convective mixing from cloud-top is likely responsible for an underestimation of the net upward water flux from the underlying moist layers. Parameterizing the cloud-top driven convection such as the convective downdrafts scheme of **Wu˙2020** is likely a parameterization development priority to further advance the representation of clouds with an intense cloud-top driven mixing layer. Such a parameterization may also help reduce the TKE at cloud-top and the time-step dependency of our ISDAC simulations (see Appendix A).'

and in the conclusion:

"However, vigorous turbulence remains unrealistically confined close to cloud top and the liquid and ice water contents are underestimated with respect to the LES simulations analyzed in **Ovchinnikov˙2014**. Those shortcomings are likely due to the absence of a parameterisation of non-local convection triggered at cloud-top in LMDZ, leading to an overly weak upward turbulent flux of water vapour below and within the cloud."

"The remaining limits of LMDZ to capture the vertical structure of the turbulence and the cloud water amounts on the ISDAC case also suggest revisiting the LMDZ shallow convection scheme to account for downward non-local mixing triggered by convective instability at the top of surface-decoupled clouds. "

---

## Author Response (AR2)

Revision of
‘Modeling the Coupled and Decoupled states of
Polar Boundary-Layer Mixed-Phase Clouds ’

Etienne Vignon, Lea Raillard al.
January 12, 2025

This document contains the response to the editorial review of ‘Modeling the Coupled and Decoupled states of Polar Boundary-Layer Mixed-Phase Clouds’ submitted to EGUSPHERE for possible publication in Atmospheric Chemistry and Physics. Comments from the Editor are in black and responses are in blue.

Comment :
« This is an interesting paper, that was evaluated slightly differently by the two expert reviewers, that were challenging the authors on both details, concepts and context. First I'd like to thank the reviewers for diligent work. Second, my judgment is that the authors have in principle responded adequately to both critique and questions, and made substantial changes to the manuscript; this work should be published - soon. I have only one minor request. »

We sincerely thank the editor for the positive comment regarding our revision work.

« This relates to the RH profiles in Figure 2, where in the original manuscript RH_liq profiles were well above 100%; not just a "few %". In most modeling, clouds would form at RH_liq < 100% to account for sub-gridscale variability assuming that pockets of air can become supersaturated at a substaurated grid volume average. In the revised manuscript this problem is fixed by taking out the RH_liq profiles and only showing RH_ice, that can easily become > 100%. I would like to see a properly calculated RH_liq profile, because that is what you can compare to the observations, and would also like a better explanation for why the originally plotted RH_liq could exceed 100%, because I did'n get the previous one and it did feel more like an excuse than an explanation. »

This is indeed a delicate point and we apologize for not being clear enough in our previous response.
In LMDZ, the state variables of the model are the specific water contents and the temperature. Relative humidity is a diagnostics variable. In convective boundary layers, the formation of liquid clouds is made through adjustment considering the saturation deficit variable $s=a*(q-q_{sat,l}(T_l))=q-q_{sat,l}(T)$, a being a thermodynamical function and $T_l$ the liquid temperature. Cloudy (liquid) air parcels thus correspond to the part of the subgrid (gaussian) distribution of s that exceeds 0.
The variable s has been chosen (see Sect. 2.3 in Jam et al. (2013)) as its distribution in convective boundary layer (observed or simulated in LES) makes it possible to distinguish the two populations or air parcels that is, those belonging to the convective updrafts and those of the environment. Moreover, and line with the beginning of your comment, clouds form when the mesh-averaged s is lower than 0 (because part of the distribution exceeds 0). However and importantly, s is a humidity variable that is not a linear function of $RH_{liq}$. This is exactly where the problem comes out as we cannot easily estimate the subgrid distribution of RHliq, knowing that of s. In other words, we assume a subgrid distribution of s, thus we know the mesh-averaged s, but we cannot properly calculate the mean RHliq.

Of course such a problem does not come out in models that do not consider a subgrid distribution of humidity (whatever the humidity variable) such as CRMs. In such models, when the cloud liquid water forms through saturation adjustment, the relative humidity wrt liquid is necessarily 100 %.

To diagnose RHliq, we thus necessarily have to make assumptions. By default (and this is what we showed in the first version of the paper), we compute a variable <RHliq> which is the ratio between the mean specific humidity [q] in the mesh and the saturation specific humidity at the mean temperature of the mesh qsat([T]). Here [] denote the mesh-average. However in convective boundary layers, q and $q_{sat}(T)$ are inversely correlated. In updrafts, air is moister and warmer compared to the environment (q is high and $q_{sat}(T)$ is high) and vice versa in the environment. Subsequently, the mean ratio if lower than the ratio of the means namely, $[RH_{liq}] < [q]/qsat[T] = <RH_{liq}>$. Hence the fact that we had <$RH_{liq}$> values exceeding 100 % in the first version of the paper.

This then raises the question of how to robustly address your comment and demonstrate a *properly calculated RH_liq profile*, given that we cannot—at least in theory—diagnose one that is consistent with the subgrid distribution of *s*. This issue was something we considered at length during the previous revision round (hence the final decision—which we agree is not fully satisfactory—to remove the RH_liq profiles).

Unfortunately, such a request is not possible with the current version of the model as we would need to modify the cloud scheme and work with subgrid distributions of RH. Note that this is what we will be doing for cirrus clouds (Borella et al. 2025) but not for boundary layer clouds. After some thoughs, we think that the most consistent way to go is to show <RHliq> in clear sky regions, and impose a value of 100 % when the mean saturation deficit is higher than 100 % wrt liquid. This is now what we show in the new version of the paper. The caption of Figure 2 has been modified accordingly. Note that the liquid layer in the simulation now corresponds to a layer where RHliq = 100 %.

Borella, A., Vignon, É., Boucher, O., Meurdesoif, Y., and Fairhead, L.: A New Prognostic Parameterization of Subgrid Ice515Supersaturation and Cirrus Clouds in the ICOLMDZ AGCM, Journal of Advances in Modeling Earth Systems, 17, e2024MS004 918,https://doi.org/10.1029/2024MS004918, e2024MS004918 2024MS004918, 2025

Jam, A., Hourdin, F., Rio, C., and Couvreux, F.: Resolved Versus Parametrized Boundary-Layer Plumes. Part III: Derivation of a Statistical Scheme for Cumulus Clouds, Boundary Layer Meteorology, 147, 421–441, 2013